# An *in vivo* strategy to counteract post-administration anticoagulant activity of azido-Warfarin

Sylvain Ursuegui[1,*], Marion Recher[1,*], Wojciech Kręzel[2,3,4,5,6] & Alain Wagner[1]

Drugs, usually long acting and metabolically stable molecules, might be the source of adverse effects triggered by complex drug interactions, anaphylaxis and drug-induced coagulopathy. To circumvent this growing drug safety issue, we herein investigate the opportunity offered by bio-orthogonal chemistry for *in vivo* drug neutralization. We design a small-molecule anticoagulant drug (Warfarin) containing an azide group that acts as a safety pin. It allows drug deactivation and restoration of physiological coagulation via *in vivo* click reaction with a suitable cyclooctyne-based neutralizing agent. In this strategy, the new molecule formed by reaction of the drug and the antidote is deprived of biological activity and prone to fast renal clearance. This 'Click & Clear' approach lays ground for new strategies in designing drugs with switchable biophysical properties.

[1] Laboratory of Functional Chemo-Systems (UMR 7199), Labex Medalis, University of Strasbourg, 74 Route du Rhin, 67401 Illkirch-Graffenstaden, France. [2] Institut de Génétique et de Biologie Moléculaire et Cellulaire, , 67404 Illkirch, France. [3] Institut de la Santé et de la Recherche Médicale, U964, 67404 Illkirch, France. [4] Centre National de la Recherche Scientifique, UMR7104, 67404 Illkirch, France. [5] Université de Strasbourg, 67404 Illkirch, France. [6] Fédération de Médecine Translationnelle de Strasbourg, 67404 Illkirch, France. *These authors contributed equally to this work. Correspondence and requests for materials should be addressed to W.K. (email: krezel@igbmc.fr) or to A.W. (email: alwag@unistra.fr).

Optimization of absorption, distribution, metabolism and excretion profile to allow the most efficient targeting of the effector site is an essential step in the course of drug development. Indeed, once the drug has been administered and after its entry into the systemic circulation, it undergoes complex distribution, metabolism and excretion processes. Consequently, the ability of the chemist to tweak molecule structure to optimize its absorption, distribution, metabolism and excretion profile without losing pharmacological activity or *in vivo* specificity accounts for a large part of a drug's therapeutic efficacy.

However, such optimized drugs, usually long acting and metabolically stable molecules, might also be the source of some adverse effects. Among classical examples of iatrogenic accidents, triggered by molecular therapy, are complex drug interactions, anaphylaxis and drug-induced coagulopathy. To circumvent this growing drug safety issue, researchers have started to develop dual drug-antidote strategies. The most successful system currently used in the clinic consists of administering a γ-cyclodextrin bearing a hydrophobic core and a hydrophilic crown tailored to scavenge the muscle relaxants rocuronium and vecuronium during general anaesthesia[1–4]. Another approach focuses on the development of a novel type of aptamer-based anticoagulant that can be neutralized with antisense oligonucleotides[5–8], porphyrines[9,10] or positively charged oligomers[11]. The system REG 1, comprising an aptamer as a reversible antagonists of coagulation factor IXa and its complementary oligonucleotides as antidote, was taken into clinic trials but, to the best of our knowledge, has been halted in phase III (ref. 12). Both of these approaches rely on specific non-covalent interaction to scavenge the exogenous compound preventing its biological effect without removing it from the organism.

To address the issue of drug neutralization and clearance, we have investigated the opportunity offered by *in vivo* bio-orthogonal chemistry. Our idea was to design a small-molecule drug containing an azide group that would act as a safety pin, allowing drug deactivation and clearance via an *in vivo* click reaction with a suitable strained cyclooctyne-containing neutralizing agent. In this strategy, the new molecule formed by reaction of the drug and the antidote should ideally be deprived of any biological activity and quickly cleared from the bloodstream via renal excretion.

Previously, click chemistry has been used to trace proteins bearing azido amino acids and lipids[13–16]. This chemistry has also been demonstrated to be efficient for labelling azidosugars in the physiologically relevant context of a mouse[17–19], and more recently for pre-targeted tumour imaging in living mice[20]. These results establish the feasibility of using *in vivo* bio-orthogonal chemical reaction in living organisms when one reactant is targeted to specific tissues (via a labelled antibody or a polymer)[21]. In pioneering work, Rossin *et al.*[22] developed a novel 2-tetrazine-functionalized clearing agents that enable the rapid reaction with and the removal of a TCO-tagged antibody from blood. This clearing agent led to 125-fold improvement of the Ab tumour-to-blood ratio 3 h after tetrazine injection. This was achieved through rapid reaction of clearing agents with the mAb in blood to remove it from the circulation and concentrate it in the liver or spleen without blocking the tumour-bound mAb.

## Results

**General**. Here we demonstrate that click chemistry can proceed in the vasculature at the level of the whole organism and can be used for targeted drug deactivation and clearance. We study the neutralization of 3-(1-(4-azidophenyl)-3-oxobutyl)-2*H*-chromen-2-one (WN$_3$, **2**), an isosteric analogue of Warfarin (W), a well-known anti-vitamin K anticoagulant agent, by click reaction with an optimized soluble analogue of BiCyclo[6.1.0] Nonyne (BCN-peg$_6$-OH **9**, Fig. 1).

**WN$_3$ (2) and clearing cycloalkyne (9) derivatives synthesis**. The introduction of an azide moiety on warfarin derivative **1a** should maintain the anticoagulant activity. Accordingly, whereas studies have shown that substitution on the 4-hydroxycoumarin motif resulted in a significant reduction of the anticoagulant activity[23], the substitution on the phenyl moiety is innocuous to the warfarin's anticoagulant activity. Indeed acenocoumarol **1b**, 4′-nitro warfarin analogue, is currently used in clinic. For this

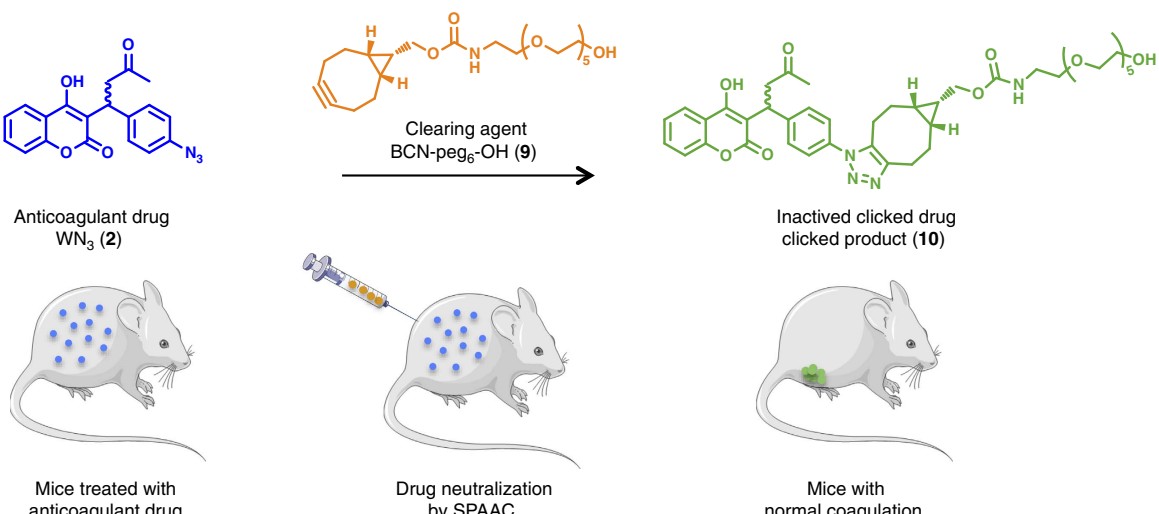

**Figure 1 | 'Click and Clear' strategy.** Biological inactivation and fast clearance of a circulating drug, Warfarin-N$_3$ **2**, by *in vivo* Strain Promoted Alkyne Azide Cycloaddition (SPAAC) reaction. A mouse submitted to anticoagulant therapy (WN$_3$, **2**) is treated with a clearing agent (BCN-peg$_6$-OH, **9**) prone to react with the anticoagulant drug. *In vivo* bio-orthogonal reaction between circulating WN$_3$ **2** and BCN-peg$_6$-OH **9** leads to the formation of an inactivated compound **10** which is readily cleared from the bloodstream, restoring normal coagulation activity. The pictures of mice and syringe have been downloaded from Servier Medical Art Database which provides these illustrations through the Creative Commons license (https://creativecommons.org/licenses/by/3.0/).

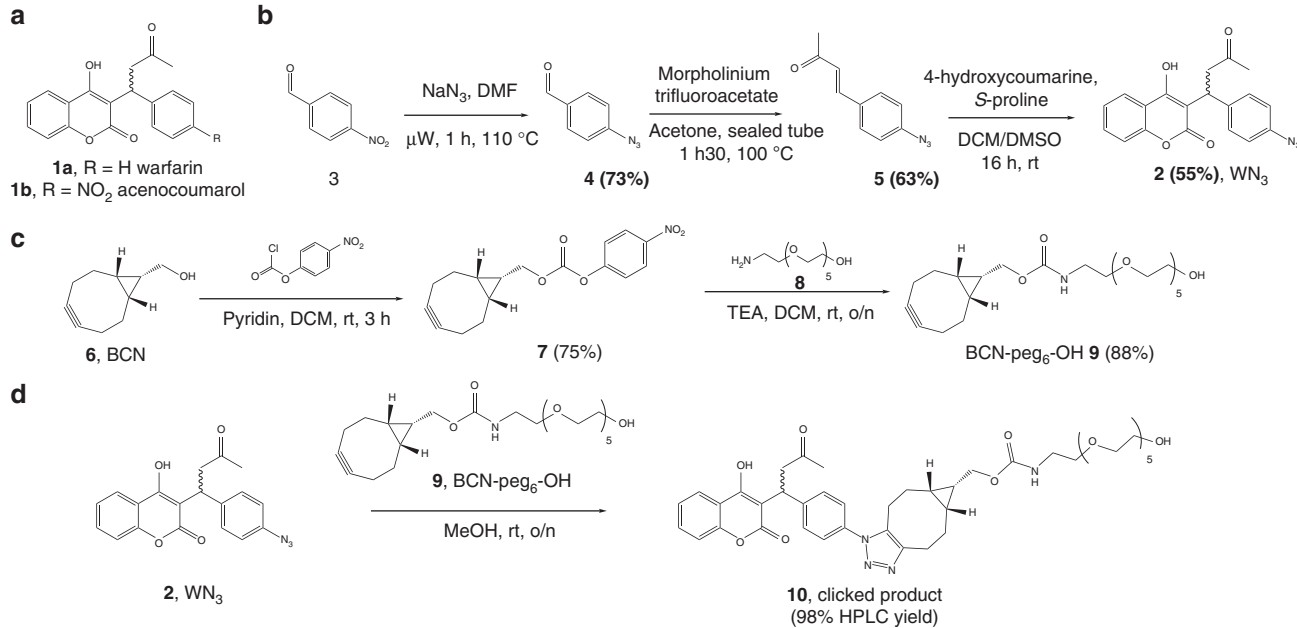

**Figure 2 | Synthesis of chemical compounds.** (**a**) Structure of Warfarin **1a** and acenocoumarol **1b** anticoagulant drugs. (**b**) Synthesis of WN$_3$ **2** anticoagulant analogue. (**c**) Synthesis of the clearing agent: BCN-peg$_6$-OH **9**. (**d**) Synthesis of the clicked product resulting from WN$_3$ **2** and BCN-peg$_6$-OH **9** reaction.

reason we decided to introduce the azide group in 4′ position of the warfarin phenyl ring. The synthesis of WN$_3$ **2** was achieved as illustrated in Fig. 2b.

To circumvent the low solubility of BCN, which could hamper its administration and activity *in vivo*, we synthesized a neutralizing agent BCN-peg$_6$-OH **9** derivative. This analogue of BCN is soluble enough to reach high concentration in the plasma and can be efficiently detected in plasma at low concentrations. The synthesis of BCN-peg$_6$-OH **9** was achieved as depicted in Fig. 2c.

The reactivity of WN$_3$ **2** was evaluated towards BCN-peg$_6$-OH **9** in plasma. Since interactions of plasma proteins with reagents could indirectly affect our measures, we first, showed that all forms of warfarin (**1a**, **2** and **10**) were completely extracted from plasma using acetonitrile—the procedure used for the subsequent kinetics studies (Supplementary Fig. 7). Then kinetic experiments were recorded in plasma using equimolar (100 μM) concentrations of both compounds. WN$_3$ **2** was found to react efficiently with BCN-peg6-OH **9** in plasma without formation of any by-product (Supplementary Fig. 11) with a rate constant of $6.8 \pm 1.8 \, \text{M}^{-1} \, \text{s}^{-1}$ (Supplementary Fig. 12).

The stability of WN$_3$ **2** and BCN-peg$_6$-OH **9** in plasma was then assessed. WN$_3$ **2** was found to be stable when kept at 37 °C in the plasma or as dried powder for more than 24 h without particular precaution. WN$_3$ **2** stability in plasma was also assessed by recording the kinetic of neutralization with BCN-peg$_6$-OH **9** after incubation of a solution of WN$_3$ **2** aged in plasma at 37 °C for 15 h. Under this condition a reaction rate constant of $5.1 \pm 0.8 \, \text{M}^{-1} \, \text{s}^{-1}$ was measured (Supplementary Fig. 13), close to that observed in the control experiment using fresh solution of reagents ($6.8 \pm 1.8 \, \text{M}^{-1} \, \text{s}^{-1}$, Supplementary Fig. 12). BCN is known to be a reactive scaffold with limited stability in biological media. We thus decided to evaluate the effect of the time of residence of BCN-peg$_6$-OH **9** in the plasma at 37 °C on its ability to neutralize WN$_3$ **2**. As expected, we found that after 15 h of incubating BCN-peg$_6$-OH **9** in human plasma at 37 °C, kinetic constant decreased with time from $6.8 \pm 1.8$ to $2.4 \pm 0.8 \, \text{M}^{-1} \, \text{s}^{-1}$ (Supplementary Fig. 14). This suggests that BCN-peg$_6$-OH **9** in

plasma shows a suitable reactivity versus stability balance during limited time-window, yet sufficiently long to be suitable for *in vivo* experiments (compare Fig. 7).

**WN$_3$ (2) anticoagulant activity.** To evaluate the anticoagulant activity of the synthesized compounds, we measured the pro-thrombin time (PT), which is the standard parameter used in clinics for the evaluation of coagulation process in patients treated with anti-vitamin K compounds. To this end, we first validated and optimized this technique for the measurement of antic-oagulant activity of pharmacological treatments in mice. In agreement with previous reports[24], the basal PT was found to range between 13 and 16 s ($14.8 \pm 0.2$ s, Supplementary Fig. 15), indicating thus that any higher PT will reflect an anticoagulant activity.

We then confirmed the activity of WN$_3$ **2** compared to the parent compound Warfarin **1a**. Two *per os* administrations of Warfarin ($10 \, \text{mg kg}^{-1}$) led to a significant increase of PT values reaching $451.2 \pm 135.1$ s ($P < 0.01$, as compared to $14.8 \pm 0.2$ in control untreated mice; Supplementary Fig. 15). Similarly, significant anticoagulant effect was also observed after *per os* administration of WN$_3$ **2** ($403.0 \pm 207.2$ s, $P < 0.05$; Supplementary Fig. 15). However, some samples in each of the two treatment groups did not coagulate within 15 min of the PT cut-off period, leading thus to high variability of experimental data and reflecting possible effect of the high dose, but also differences in absorption and metabolism of both compounds. To reduce such variability, we focused on WN$_3$ **2** and used lower doses and a different administration method. To this end, mice were injected (intraperitoneal (IP)) twice at a 24-h interval with different doses of WN$_3$ **2** and were killed 6 h after the last injection. We observed a dose-dependent increase of PT reaching $25.9 \pm 2.8$ s for $1 \, \text{mg kg}^{-1}$ and $58.5 \pm 9.7$ for $2 \, \text{mg kg}^{-1}$ (Fig. 3a). This dose/response effect displayed low variability and was supported by significant effect of the dose ($F[2,15] = 11.8$, $P < 0.001$; one-way analysis of variance (ANOVA)). In the next round of experiments, we evaluated optimal delay after the last

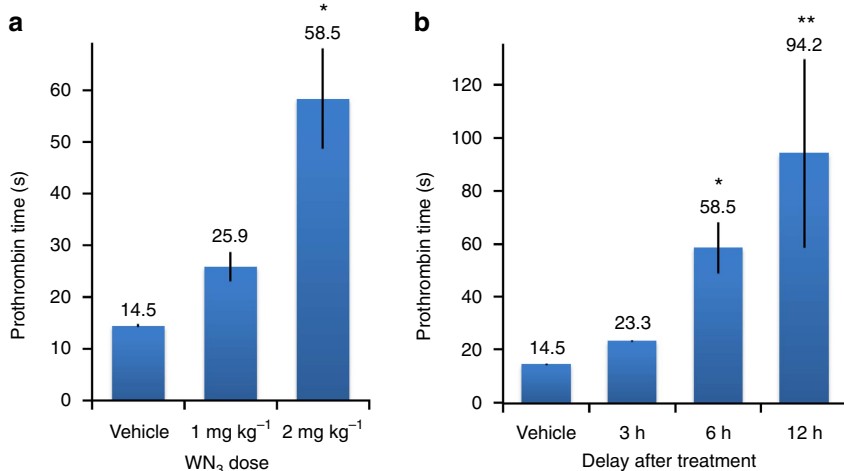

**Figure 3 | *In vivo* biological activity of WN$_3$ (2) analogue.** (**a**) PT depends from the WN$_3$ **2** dose delivered by IP injection. Dose/response curve of WN$_3$ **2** was measured 6 h after the second injection of 1 and 2 mg kg$^{-1}$ of WN$_3$ **2** and compared to vehicle-treated mice (two IP injections, $n=4$). (**b**) PT depends on the delay after treatment. WN$_3$ **2** (2 mg kg$^{-1}$) was administered twice with a 24-h interval by IP injection and PT was measured 3, 6 and 12 h ($n=4$–6/group) after the second treatment and compared to vehicle-treated mice. Statistical differences identified with protected least significant difference (PLSD) Fischer test were indicated as: *$P<0.05$, **$P<0.01$ as compared with vehicle-treated mice. Error bars indicate s.e.m. values.

treatment to obtain significant and reproducible anticoagulant effect. After two IP injections of 2 mg kg$^{-1}$ of WN$_3$ **2**, mice were killed at different time points for blood collection and PT measurements. A time dependence of anticoagulant activity was supported by significant effect of the test time after injection on PT measurements ($F[3,16]=4.8$, $P<0.05$; one-way ANOVA). 3 h after the last treatment, a slight but not significant increase of PT was observed (23.3 ± 0.3 s as compared to 14.5 ± 0.5 s in control group; $P=0.7$). This increase was enhanced and significant at 6 h (58.5 ± 9.7; $P<0.05$) and 12 h (94.2 ± 35.6 s; $P<0.01$), reflecting clearly a time-dependent activity of WN$_3$ **2** (Fig. 3b). Although 12 h after treatment, biological activity was the highest, it was associated with an important biological variability reflected by a high s.e.m. As a result of these analyses, we retained for further functional studies an IP injection of 2 mg kg$^{-1}$ and PT measurement after 6 h as optimal experimental conditions to investigate *in vivo* drug inactivation, providing significant, ∼3.5-fold increase of anticoagulant activity (as compared to the control group) with low variability. Moreover, this dose of WN$_3$ **2** corresponds to doses of Warfarin **1a** used in clinical conditions as Warfarin dose used in human varies to attain 2–4-fold increase of PT. We also confirmed that the prolongation of PT after WN$_3$ **2** or Warfarin **1a** treatment results in the concomitant decrease of the three-major vitamin K-dependent clotting factors, II, VII and X as revealed by PT measures using dedicated clotting factor assays (Supplementary Figs 19 and 20).

**Cycloalkyne (9) and clicked product (10) biological activities.** Prior to testing the WN$_3$ **2** inactivation by click reaction *in vivo*, we investigated whether SPAAC modification of WN$_3$ carried out *in vitro* affects anticoagulant activity of WN$_3$ **2**. The SPAAC modified WN$_3$ **10** did not display any anticoagulant activity (12.7 s ± 0.4; $P=0.6$, as compared to 14.5 ± 0.5 s in IP vehicle treatment, Fig. 4f) when administered by two IP injections at the dose of 4.6 mg kg$^{-1}$ corresponding to 1 equimolar quantity of WN$_3$ at 2 mg kg$^{-1}$. Importantly, an IP or sub-cutaneous (SC) injection of BCN-peg$_6$-OH **9** (26 mg kg$^{-1}$ corresponding to 10 equimolar quantity of 2 mg kg$^{-1}$ of WN$_3$), did not affect anticoagulant activity in mice (IP: 12.0 ± 0.4 s, $P>0.2$ and SC: 13.0 ± 0.7 s, $P>0.9$ as compared to vehicle-treated mice). Further increasing the BCN-peg$_6$-OH **9** dose to 52 mg kg$^{-1}$,

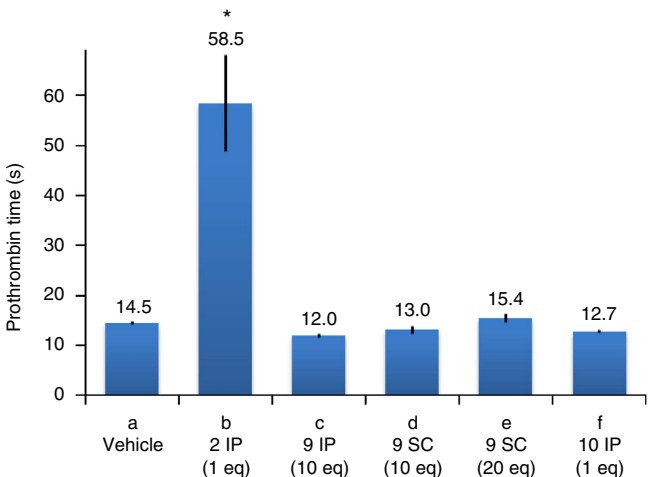

**Figure 4 | Anticoagulant activity of BCN-peg$_6$-OH (9) and *in vitro*-clicked product (10).** PT values were determined for different compounds following two IP or SC injections with a 24-h interval, as follows: (a) vehicle IP injections to establish control group of mice, $n=4$. (b) WN$_3$ **2** IP injections (2 mg kg$^{-1}$, $n=7$). (c) BCN-peg$_6$-OH **9** IP injections (26 mg kg$^{-1}$ (10 eq), $n=4$). (d) BCN-peg$_6$-OH **9** SC injections (26 mg kg$^{-1}$ (10 eq), $n=4$). (e) BCN-peg$_6$-OH **9** SC injections (52 mg kg$^{-1}$ (20 eq), $n=4$). (f) Clicked product **10** IP injections (4.6 mg kg$^{-1}$ (1 eq), $n=4$). *$P<0.1$, PLSD Fisher comparison with vehicle-treated mice. Error bars indicate s.e.m. values.

corresponding to 20 equimolar quantity of WN$_3$ **2** at 2 mg kg$^{-1}$, did not alter blood coagulation as analysed for SC administration (15.3 ± 0.8, $P=0.9$). Such observations indicate that if click reaction proceeds *in vivo* it should induce the inactivation of WN$_3$ **2** anticoagulant activity and the decrease of the PT.

As all measures of BCN-peg$_6$-OH **9** activity were performed in non-stimulated conditions when PT values are naturally low, we could not exclude the possibility that some endogenous pro-coagulant activity of BCN-peg$_6$-OH **9** would be masked by the 'floor effect' of PT measurements. To address this point, we tested activity of BCN-peg$_6$-OH **9** in mice in which PT was increased by treatment with Warfarin **1a** (which does not bear an azide group) and thus which should not be sensitive to the click reaction. As

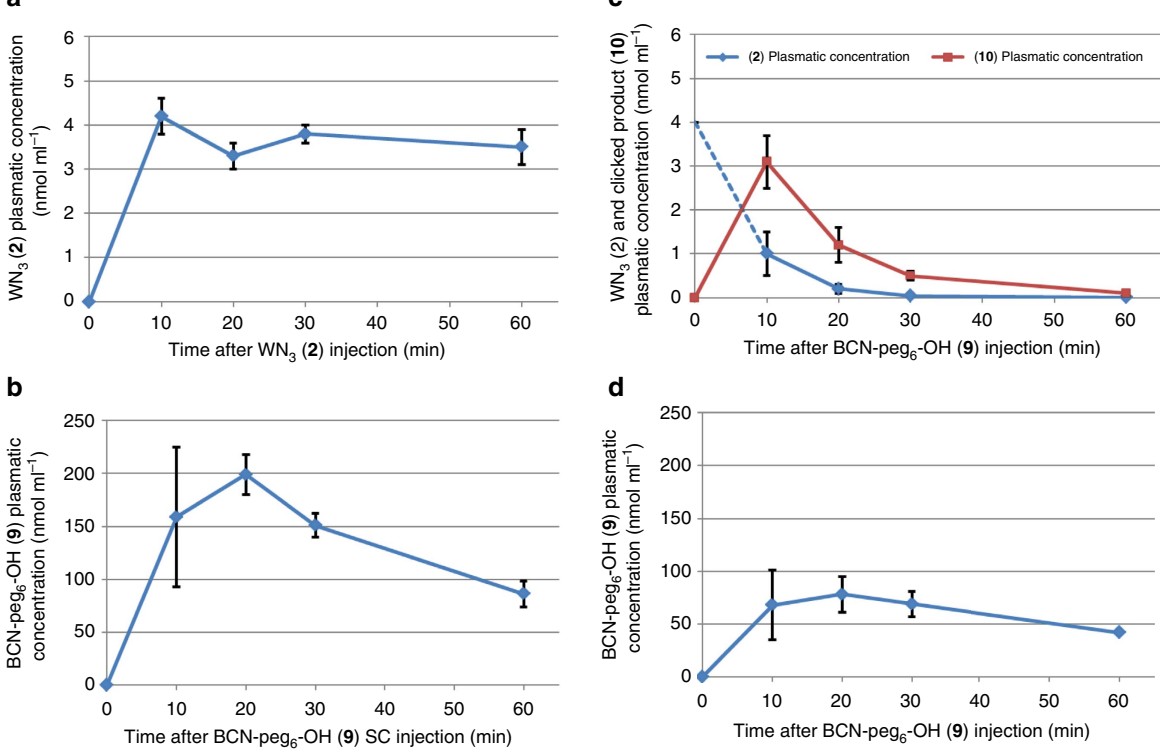

**Figure 5 | LC–MS/MS analysis in plasma. (a,b)** WN$_3$ (**2**) and BCN-peg$_6$-OH (**9**) pharmacokinetics. (**a**) Plasma concentration of WN$_3$ **2** after IP injection (2 mg kg$^{-1}$, $n = 4$). (**b**) Plasmatic concentration of BCN-peg$_6$-OH **9** after SC injection (52 mg kg$^{-1}$, 114.4 µmol kg$^{-1}$, $n = 4$). (**c,d**) LC–MS/MS analysis of *in vivo* click reaction. WN$_3$ **2** (2.0 mg kg$^{-1}$, 5.72 µmol kg$^{-1}$, 1 eq) was administered by IP injection followed by the BCN-peg$_6$-OH **9** delivery (52 mg kg$^{-1}$, 114.4 µmol kg$^{-1}$, 20 eq) by SC injection 5 min later. Blood collection was performed after BCN-peg$_6$-OH **9** injection to determine WN$_3$ **2**, BCN-peg$_6$-OH **9** and clicked product **10** concentration by LC–MS/MS analysis. (**c**) Blue: plasma concentration of WN$_3$ **2**. Red: plasma concentration of clicked product **10**. (**d**) Plasma concentration of BCN-peg$_6$-OH **9**. These experiments were performed in quadruplicate and error bars indicate s.d.

expected, using the same protocol of injections as for WN$_3$ **2** we did not observe any reduction of PT in mice injected with 2 mg kg$^{-1}$ of Warfarin **1a** followed by 20 eq of BCN-peg$_6$-OH **9**, indicating clearly that BCN-peg$_6$-OH **9** on its own does not display any pro-coagulant activity (Supplementary Fig. 16). In line with this observation, BCN-peg$_6$-OH **9** did not display any major effect on Warfarin **1a** mediated reduction of clotting factors (Supplementary Fig. 19).

**Pharmacokinetics in plasma.** We hypothesized that to perform an *in vivo* reaction the critical point would be the use of a chemical antidote possessing a similar pharmacokinetic profile as the targeted drug. For this reason, pharmacokinetics of WN$_3$ **2** and BCN-peg$_6$-OH **9** in living mice were determined using liquid chromatography–mass spectrometry (LC–MS)/MS analysis. WN$_3$ **2** (2 mg kg$^{-1}$) was administered by IP and BCN-peg$_6$-OH **9** was delivered by SC injection (52 mg kg$^{-1}$ corresponding to 20 equimolar quantity with respect to 2 mg kg$^{-1}$ of WN$_3$). Blood samples were collected from the tail vein at selected time intervals (10, 20, 30 and 60 min) and WN$_3$ **2** and BCN-peg$_6$-OH **9** levels were quantified in plasma preparations using LC–MS/MS analysis. Using this procedure, the maximal plasma concentration of WN$_3$ **2** was observed at 10 min (4.2 ± 0.4 nmol ml$^{-1}$) and remained relatively constant for 1 h (Fig. 5a). This long circulation time may reflect strong binding of WN$_3$ **2** to plasma proteins, which was comparable to that observed to non-modified warfarin **1a** (97%, Supplementary Figs 24 and 25). For BCN-peg$_6$-OH **9**, the plasma concentration was the highest between 10 and 30 min after SC injection, reaching ∼200 nmol ml$^{-1}$ and decreasing

more rapidly as compared to WN$_3$ **2** (Fig. 5b). Thus, due to WN$_3$ **2** fast uptake into the bloodstream, we decided to inject BCN-peg$_6$-OH **9** 5 min after WN$_3$ **2** administration to maximize the time during which the two substrates would be present in the blood at high concentrations.

To address the *in vivo* formation of the clicked product **10,** we first carried out a series of LC–MS/MS analysis of plasma samples from mice following WN$_3$ **2** and BCN-peg$_6$-OH **9** treatments. To guarantee efficient *in vivo* reaction, we used 20 equivalents of BCN-peg$_6$-OH **9** with respect to 2 mg kg$^{-1}$ of WN$_3$. WN$_3$ **2** (2.0 mg kg$^{-1}$ corresponding to 5.7 µmol kg$^{-1}$, 1 eq) was administered by IP injection whereas BCN-peg$_6$-OH **9** (52 mg kg$^{-1}$, 114.4 µmol kg$^{-1}$, 20 eq) was injected SCly 5 min later, to attain simultaneous maximal WN$_3$ **2** and BCN-peg$_6$-OH **9** plasma concentration. By choosing a different time point and route of administration for BCN-peg$_6$-OH **9**, we aimed to prevent direct contact and reaction between the compounds in the abdomen thereby favoring drug inactivation in the bloodstream. Plasma samples were collected at 10, 20, 30 and 60 min after the administration of BCN-peg$_6$-OH **9**. The clicked product **10** was already detected at a concentration of 3.1 ± 0.6 nmol ml$^{-1}$ after 10 min, indicating that the reaction took place *in vivo* (Fig. 5c). Furthermore, inactivated drug **10** appeared to be the sole product arising from WN$_3$ **2** since the sum of the concentration of the remaining WN$_3$ **2** (1.0 ± 0.5 nmol ml$^{-1}$, Fig. 5c) and the clicked product **10** (3.1 ± 0.6 nmol ml$^{-1}$) corresponds to maximal plasma concentration of WN$_3$ **2** expected in absence of BCN-peg$_6$-OH **9** (4.2 ± 0.4 nmol ml$^{-1}$; compare Fig. 5a). Accordingly, the concentration of circulating BCN-peg$_6$-OH **9** was reduced (compare Fig. 5b,d). Most notably, these results also revealed that

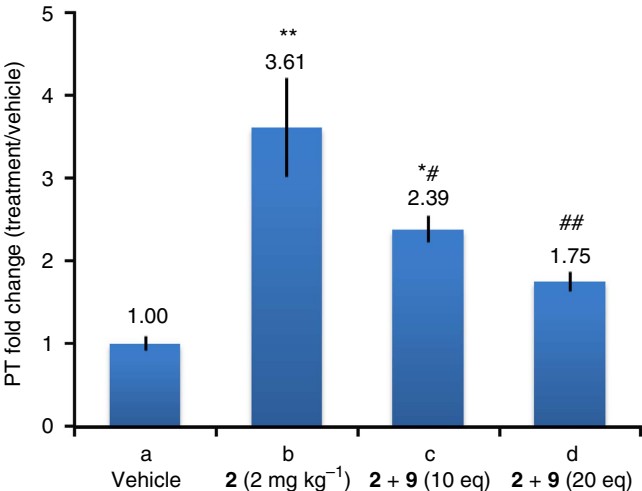

**Figure 6 | WN₃ (2) biological activity after *in vivo* click reaction.** Anticoagulant activity represented as a ratio of PT measured for vehicle IP injections. (a) Vehicle IP injections to establish control group of mice ($n = 6$) and used as reference (1.00). (b) Animal treated only with WN₃ **2** (2.0 mg kg$^{-1}$, 5.7 μmol kg$^{-1}$, 1 eq) twice with a 24-h gap by IP injection ($n = 7$). (c) WN₃ 2 IP injections (2.0 mg kg$^{-1}$, 5.7 μmol kg$^{-1}$, 1 eq) were followed by SC injections of BCN-peg₆-OH **9** (26 mg kg$^{-1}$, 57.2 μmol kg$^{-1}$, 10 eq) 5 min post-WN₃ **2** delivery ($n = 4$). (d) WN₃ 2 IP injections (2.0 mg kg$^{-1}$, 5.7 μmol kg$^{-1}$, 1 eq) were followed by SC injections of BCN-peg₆-OH 9 (52 mg kg$^{-1}$, 114.4 μmol kg$^{-1}$, 20 eq) 5 min post-WN₃ 2 delivery ($n = 7$). Statistical differences identified with PLSD Fischer test were indicated as: *$P < 0.05$ and **$P < 0.01$, as compared to vehicle-treated mice and #$P < 0.05$ and ##$P < 0.01$, with respect to WN₃ **2** treated mice. Error bars indicate s.e.m. values.

the click reaction between **2** and **9** seemed to trigger quick drug elimination because there was no more free WN₃ **2** nor click product **10** in plasma 60 min after administration of BCN-peg₆-OH **9** (Fig. 5c). This is consistent with the pharmacokinetic profile of BCN-peg₆-OH **9** for which the plasma concentration diminished faster after reaching its peak than does WN₃ **2** (Fig. 5a,b).

**WN₃ (2) inactivation by click *in vivo*.** Our data indicate that: (i) clicked product **10** has no anticoagulant activity and (ii) *in vivo* click reaction can proceed efficiently and lead to the disappearance of WN₃ **2** in the plasma. To address the functional relevance of WN₃ **2** modification by SPAAC *in vivo*, we studied the anticoagulant activity after two IP injections of WN₃ **2** (2 mg kg$^{-1}$) with a 24-h interval and each followed 5 min later by a SC administration of 10 or 20 equivalents of BCN-peg₆-OH **9** (26 or 52 mg kg$^{-1}$ respectively). The analysis of PT performed on plasma samples 6 h after the last injection revealed that WN₃ **2** displayed different anticoagulant activity depending on the presence or not of BCN-peg₆-OH **9** which was reflected by significant interaction between effects of WN₃ **2** and BCN-peg₆-OH **9** administration ($F[2,29] = 3.8$, $P < 0.05$; two-way ANOVA, Fig. 6). The neutralizing effect of BCN-peg₆-OH **9** was dose-dependent ($F[2,15] = 5.0$, $P < 0.05$; one-way ANOVA for BCN-peg₆-OH **9** activity in WN₃ **2** treated mice) as 10 equivalent of BCN-peg₆-OH **9** reduced WN₃ activity (from 3.61 to 2.39 PT fold change), whereas the reduction was even stronger after an injection of 20 equivalent of BCN-peg₆-OH **9** (from 3.61 to 1.75 PT fold change). Importantly, injection of 20 eq of BCN-peg₆-OH **9** normalized PT in WN₃ **2** pre-treated mice ($P = 0.1$ not significant as compared to 20 eq of BCN-peg₆-OH **9** or vehicle

injected group), whereas PT observed after similar application of only 10 eq of BCN-peg₆-OH **9** still remained higher than the corresponding control groups ($P < 0.05$, as compared to 10 eq of BCN-peg₆-OH **9** or vehicle group). The significant decrease of the WN₃ **2** activity following BCN-peg₆-OH **9** treatment was the consequence of the *in vivo* click reaction and did not result from pro-coagulant activity of BCN-peg₆-OH **9**, as two vehicle or warfarin IP treatments followed by administration of BCN-peg₆-OH **9** at 26 or 52 mg kg$^{-1}$ did not alter PT measures ($P > 0.2$ and $P > 0.9$ for 10 or 20 eq of BCN-peg₆-OH **9** groups as compared to vehicle treatment group, Fig. 4 and Supplementary Fig. 16). The pro-coagulant effects of BCN-peg₆-OH **9** were comparable to activity of high dose (20 mg kg$^{-1}$) of vitamin K1 in WN₃ **2** treated animals with respect to both reduction of PT values (Supplementary Fig. 17) and decreased activity of all three clotting factors, II, VII and X (Supplementary Fig. 20).

BCN-peg₆-OH **9** administration delayed by 10 and 30 min can also inactivate WN₃ **2** with equal efficiency as BCN-peg₆-OH **9** administration at 5 min post-WN₃ **2** treatment (Supplementary Fig. 17). This observation is important as BCN applied at 30 min should attain its maximal levels in the organism 10–20 min later, which corresponds to 40–50 min after WN₃ **2** administration. We found that at this time point WN₃ **2** is present not only in plasma but also in target organs such as the liver (Fig. 7), suggesting that BCN-peg₆-OH **9** may react with WN₃ **2** and clear it not only from circulation, but also from target organs.

However, the PT measures after neutralization of WN₃ **2** by BCN-peg₆-OH **9** showed a tendency ($P = 0.1$) to stay above the PT of control mice, which could be due to re-equilibration of WN₃ **2** into circulation from other tissues such as the liver. To address this important point, we have measured the amount of WN₃ **2** remaining in the liver after BCN-peg₆-OH **9** treatment. Accordingly, at 45 min after BCN-peg₆-OH **9** treatment, there are low but detectable levels of free WN₃ **2** in the liver; 4.3 nmol g$^{-1}$ as compared to 31.4 nmol g$^{-1}$ observed at the same time point in mice injected with WN₃ **2** alone (compare 'no click reaction' and '*in vivo* click reaction' panels in Fig. 7). These data suggest that the remaining low amount of WN₃ **2** in the liver after the click reaction account at least in part for the residual anticoagulant activity. However, we cannot exclude re-equilibration of WN₃ **2** from another compartment. Indeed, the neutralizing strained alkyne was designed to have a fast clearance preventing later reaction with re-equilibrated WN₃ **2**. We thus measured the renal clearance of WN₃ **2**, BCN-peg₆-OH **9** and clicked product **10**.

**Renal clearance of WN₃ (2) after *in vivo* click reaction.** To study the fate of WN₃ **2** following BCN-peg₆-OH **9** administration, we compared its distribution in three body compartments (plasma, liver and urine) at 50–60 min after single IP administration of WN₃ **2** (2 mg kg$^{-1}$) in mice which received BCN-peg₆-OH **9** (single SC injection of 52 mg kg$^{-1}$, 20 eq) or vehicle at 5 min after WN₃ **2** treatment. Two-way ANOVA analyses revealed that WN₃ **2** distribution in different compartments was significantly different depending on the application of BCN-peg₆-OH **9** as supported by significant interaction between body compartment and BCN-peg₆-OH **9** application ($F[2,13] = 6.27$, $P < 0.05$). This difference reflects almost complete absence of free WN₃ **2** in plasma, liver and urine after BCN-peg₆-OH **9** treatment (Fig. 7). This dramatic reduction of WN₃ **2** was at the expense of a strong increase of clicked product **10**, which was found predominantly in the urine (95.3% of total clicked product detected in all three body compartments, respectively). These data indicate rapid renal clearance of clicked WN₃ **10** which might be due to strong endogenous preference for such mode of clearance of unbound BCN-peg₆-OH **9** as indicated by almost exclusive

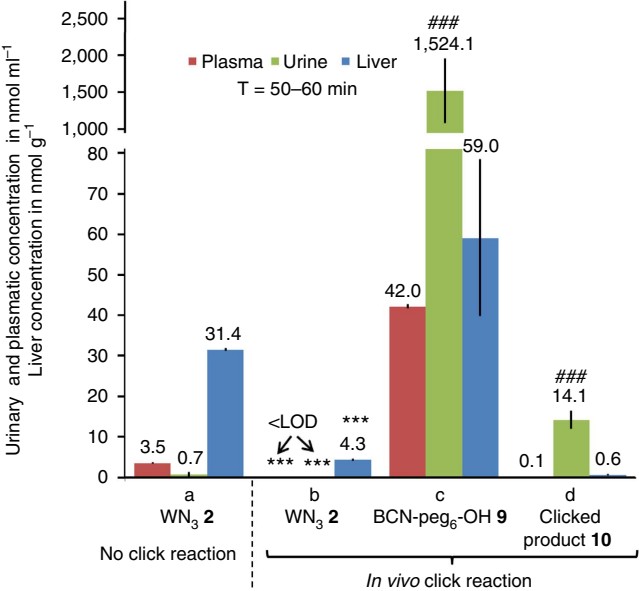

**Figure 7 | Effects of *in vivo* click reaction on WN₃ (2) distribution and renal excretion.** (a) WN$_3$ **2** concentration was measured in the plasma, urine and liver of mice treated with a single dose of WN$_3$ **2** (2.0 mg kg$^{-1}$, 5.7 μmol kg$^{-1}$, 1 eq) by IP injection ($n = 3$). (b–d) Analyses of free and clicked WN$_3$ **2** and BCN-peg$_6$-OH **9** after single IP injection of WN$_3$ **2** (2.0 mg kg$^{-1}$, 5.7 μmol kg$^{-1}$, 1 eq) followed 5 min later by SC injections of BCN-peg$_6$-OH **9** (52 mg kg$^{-1}$, 114.4 μmol kg$^{-1}$, 20 eq) ($n = 3$). (a–d) Samples collection have been performed 50–60 min after the last injection. Note that plasma (red) and urine (blue) concentrations of all substrates were reported in nmol ml$^{-1}$ whereas liver (green) levels were shown in nmol g$^{-1}$. <LOD (limit of detection), below the detection limit. ***$P < 0.001$ as compared to WN$_3$ **2** in respective compartments shown in 'No click reaction' panel. ###$P < 0.001$ as compared to liver and plasma measures of the corresponding compound. Error bars indicate s.e.m.

localization of free BCN-peg$_6$-OH **9** in the urine (93.8% of all BCN-peg$_6$-OH **9** found in all three body compartments, Fig. 7). The data support the possibility that the efficiency of the click reaction and elimination of clicked product is limited by the time-window of the availability of BCN due to its fast renal clearance thus preventing neutralization of re-equilibrated WN$_3$ **2**.

In summary, to investigate the opportunity offered by bio-orthogonal chemistry for an *in vivo* drug neutralization, we designed and synthesized a bioactive analogue of Warfarin **1a** bearing an azide (WN$_3$ **2**) and a neutralizing/clearing agent bearing a strained alkyne (BCN-peg$_6$-OH **9**). We confirmed that the product arising from the reaction between WN$_3$ **2** and BCN-peg$_6$-OH **9** did not have anticoagulant property. Pharmacokinetics of WN$_3$ **2** and BCN-peg$_6$-OH **9** were measured to determine the optimum administration conditions for both compounds. Finally, we showed that the treatment of a mouse submitted to anticoagulant therapy (WN$_3$ **2**) with BCN-peg$_6$-OH **9** leads to the *in vivo* formation of the inactivated compound **10** and total elimination of plasma WN$_3$ **2** via renal clearance as demonstrated by LC–MS/MS analysis. Measurement of the PT revealed the concomitant restoration of normal anticoagulant activity. This 'Click & Clear' strategy opens interesting prospects not only for the development of new drugs with switchable biophysical properties, but also for the design of a universal reversal agent with adequate pharmacokinetic/biodistribution properties that could neutralize any azide-labelled drug.

## Methods

**Animals and *in vivo* experiments.** Young adult C57BL6N male mice (Charles River, France) at 8–12 weeks of age were used throughout this study. All mice were housed in 07:00–19:00 hours light/dark cycle in individually ventilated cages (Tecniplast, Italy). Food and water were freely available. All experiments were approved by local ethics committee and carried out in accordance with the European Community Council Directives of 24 November 1986 (86/609/EEC) and in compliance with the guidelines of CNRS and the French Agricultural and Forestry Ministry (decree 87848).

**Formulation of warfarin and BCN derivatives.** Warfarin derivatives (Warfarin **1**, WN$_3$ **2** and clicked product **10**) and BCN-peg$_6$-OH **9** were solubilized in a vehicle solution of 10% (w/w) CDX in PBS and used at 2 ml kg$^{-1}$ with exception of *per os* treatments for which Warfarin **1a** or WN$_3$ **2** were suspended in 0.5% methylcellulose solution and used at 10 ml kg$^{-1}$. In brief, for functional studies, Warfarin **1a** or WN$_3$ **2** were administrated *per os* or by IP injection twice with a 24-h interval and mice were killed for blood collection at different time points after the last injection, as indicated. For WN$_3$ **2** inactivation and the corresponding control experiments, 10 or 20 equivalents (with respect to WN$_3$ **2** quantity) of BCN-peg$_6$-OH **9** or vehicle were injected subcutaneously 5 min after each of the two WN$_3$ **2** or vehicle IP treatments and blood was collected 6 h after the last injection for prothrombin analysis. All treatments were carried out in the morning between 07:00–09:00 hours. To facilitate comparisons between different experimental conditions, changes of PT measures were reported as fold change with respect to PT values in control mice, and were calculated as the mean of ratios of each individual PT value from a specific treatment group to the mean PT value of control group.

**LC–MS/MS analysis.** For pharmacokinetic studies, one treatment was performed followed by repeated blood samples collections from the tail vein at selected time intervals (10, 20, 30 and 60 min) using heparin-coated capillaries. Plasma samples were prepared by centrifugation at room temperature at a minimum of 10 min after blood collection. This minimal incubation period aimed to reduce an error which could occur at very short periods of incubation and in consequence incomplete plasma preparation. To each 20 μl sample of of plasma, 30 μl of acetonitrile was added and the samples were centrifuged for 7 min at 4,000 r.p.m. in order to eliminate proteins. Supernatants were then analysed by LC–MS/MS (Supplementary Fig. 21). For LC–MS/MS analysis of *in vivo* click reaction, WN$_3$ **2** delivery by IP injection was followed 5 min later by SC injection of BCN-peg$_6$-OH **9**.

**Evaluation of prothrombin time and clotting factors.** To evaluate the PT, mice were euthanised by increasing dose of CO$_2$ and blood was collected by intra-cardiac puncture immediately after respiratory arrest. Sodium citrate was added immediately for a final concentration of 1% and plasma was prepared by centrifugation at room temperature a minimum of 10 min after blood collection. PT was measured using a Start4 benchtop hemostasis analyzer and dedicated kit (Neoplastine, Diagnostica Stago) according to manufacture instructions. To evaluate clotting factors we used dedicated kits (Diagnostica Stago). The calibration of the system was performed at the beginning of each analysis using reference plasma samples (Coag Control N + P, Diagnostica Stago) according to manufactures instructions (Supplementary Fig. 18).

**Data availability.** Data supporting the findings of this study are available within the article and its Supplementary Information files and from the corresponding author upon reasonable request.

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

## Acknowledgements

Financial support was provided by the International Center for Frontier Research in Chemistry (icFRC) and the Agence Nationale pour la recherche (ANR). This study was also supported by the grant ANR-10-LABX-0030-INRT, a French State fund managed by the Agence Nationale de la Recherche under the frame program Investissements d'Avenir ANR-10-IDEX-0002-02. We thank Michel Mosser (LFCS-UMR7199) for BCN synthesis, Patrick Gizzi (PCBIS-TechMed) for LC–MS/MS analysis, Nelly Frossard and François Daubeuf (UMR 7200-Laboratoire d'innovation thérapeutique) for helpful discussions of *in vivo* experiments and formulation for drug treatments. We also gratefully acknowledge Prof. Andrew Griffiths (Laboratory of Biochemistry, ESPCI Paris) for proofreading our manuscript and Dr Jean-Pierre Rameau (Hôpital de Hautepierre) for helpful discussions.

## Author contributions

A.W. and W.K. conceived and designed the project study. M.R. and S.U. synthesized and characterized the molecules studied in this work. W.K., M.R. and S.U. performed the *in vivo* experiments and/or evaluated anticoagulant activity. A.W., W.K. and S.U. wrote the manuscript.

## Additional information

**Competing interests:** The authors declare no competing financial interests.

**Publisher's note**: 

