## [Peer Review File · Nature Communications]

Reviewers' comments:

Reviewer #1 (Remarks to the Author):

Wagner and colleagues describe an appealing approach to modulate the properties of a drug, in this case warfarin, by means of click chemistry in vivo. To temporally control the extent of warfarin activity they modified it with an azide and were able to rapidly deactivate and clear the active drug by administering a cyclooctyne agent. As a result the anticoagulation activity was reduced ca 50 %. The work has a potential impact on the way certain long lasting drugs can be designed. The experimental approach and data presentation is good.

In my opinion this paper can become suitable for publication, provided below issues are adequately addressed.

In the introduction the authors state:

" While these results establish the feasibility of using in vivo bioorthogonal chemical reaction in living organism when one reactant is targeted to specific tissues (via a labeled antibody or a polymer), the possibility of neutralizing a circulating drug that requires bioorthogonal reactions to proceed at the level of the whole organism still remained unknown."

The authors have omitted the work from Rossin et al in J. Nucl. Med. 2013, 54, 11, 1989-1995. Clearing agents based on tetrazine-modified albumin or microparticles were used to react (click) with and rapidly clear freely circulating trans-cyclooctene-modified antibody from blood in mice. In addition to including this work in the introduction, the paper will be stronger if the authors compare their click&clear design and corresponding results to the other published click&clear design.

The authors observed a ca 10-fold increased reactivity in serum compared to PBS and suggest that this could be due to hydrophobic interactions between BCN and serum proteins. Usually these click reactions are accelerated when going from organic solvents to water, which does not seem to fit with above explanation. In addition such interactions surely would decrease the availability of the BCN for the azide ?

As a side note, the authors should note the concentrations used in this experiment to accompany the statement regarding the complete reaction yield in 15 minutes.

The authors achieve effective deactivation/clearing of warfarin (Figure 5), but the effect on anticoagulation appears to be modest: ca 2 fold. Could this be due to re-equilibration of warfarin from other tissues into circulation after the warfarin the blood has cleared ? The studied time frame in Figure 5 is 1 h while the results in Figure 6 are after 6 hours, so possibly some re-equilibration could have occurred?

With respect to the previous point and in general, quite some background information is missing:

- what is known about warfarin PKPD ? should it indeed re-equilibrate from other tissues back into the blood ?

- Going back to the previously quoted sentence in the introduction:

" the possibility of neutralizing a circulating drug that requires bioorthogonal reactions to proceed at the level of the whole organism still remained unknown."

Does this mean that the authors expect that their BCN agent extravasates and can deactivate warfarin reservoirs outside the vasculature ?

- What is the reason for the very slow clearance of warfarin and their azido-warfarin? could it be due to interaction with albumin for example ?

- Do the used warfarin concentrations correspond to clinical concentrations (when accounted for mouse vs man) ?
- Is a 50 % reduction of anti-coagulation activity clinically relevant? What is the clinic looking for in this respect ?
- How does the efficacy of their approach compare to those referenced in the introduction ?

Minor comments:

- the last line of the second paragraph in the introduction should provide one or more references.
- faulty grammar in the last line of paragraph 4 of the introduction
- BCN-oeg6-OH should be BCN-peg6-OH
- section "Click in vivo, metabolomic analysis": line 5 is unclear. Amongst others by confusing use of "addition". Perhaps better to use "the sum".
- section "WN3 (2) inactivation by click in vivo": the term "PT fold change" is unclear and also faulty. Consider changing into for example "PT ratio"
- section Methods, Metabolomic analysis: should "minimum 10 min" be "maximum 10 min"?
- Metabolomic should be Metabolic

Reviewer #2 (Remarks to the Author):

Wagner and colleagues describe a "click & clear" approach to control the anticoagulant activity of warfarin analog. The authors synthesized an azido warfarin derivative and characterized its activity in vivo. The azido drug could be covalently targeted with a complementary cyclooctyne; the "clicked" conjugate harbored no anticoagulant properties. Thus, the bioorthogonal ligation could effectively turn off warfarin activity. While this is an interesting concept, I have some significant concerns about the experiments and overall approach (see below). I do not recommend publication at this time.

1. The mechanism of anti-coagulant activity in vivo is still a bit nebulous. The authors demonstrate that the cyclooctyne alone can impact anti-coagulant activity (Figure 4). The data in Figure 6 are incredibly difficult to interpret. How is it possible to rule out that the cyclooctyne itself (and not the "clicked" conjugate) is responsible for the PT differences? Perhaps more rigorous competition experiments are necessary.
2. The authors do not make a strong case for the "clickable" antidote. One could argue that the effects of warfarin could be more easily tuned using other small molecule inhibitors, etc. that target other relevant pathways. The "click" reaction could be an unnecessary complication. (Note that large concentrations of these reagents are necessary for efficient in vivo ligations.)
3. I had a difficult time assessing the quality of some experiments based on the few details provided. The PT assay itself was not well described. Sample data/spectra for key LCMS studies were also not provided. Such studies are notoriously difficult to perform in the absence of radiolabels, etc. Additionally, no NMR spectra were provided to assess purity.
4. Aryl azides and cyclooctynes are not completely stable/inocuous. (Aryl azides are also difficult to characterize via conventional HPLC/LCMS owing to photolysis issues; thus, the lack of data noted in point #3 above are all the more important to include.) Cyclooctynes are also known to be sequestered by numerous serum proteins and membranes. Did the authors observe such phenomena? (Such studies might require radiolabels or some other identifiable tag). Off-target labeling could be detrimental to in vivo systems.

Reviewer #3 (Remarks to the Author):

To evaluate the potential clinical utility of copper-free click chemistry, Ursuegui and colleagues synthesized an azido-modified derivative of warfarin (designated WN3) and examined the capacity of a cyclooctyne reagent to neutralize its anticoagulant activity in mice. The authors show that (a) WN3 prolongs the prothrombin time (PT) in a time- and dose-dependent manner, and (b) cyclooctyne reagent, which they designate BCN-oeg6-ON (BCN) reverses the PT prolongation produced by WN3. Based on these findings, they conclude that this "click and clear" strategy provides a framework for developing reversible drugs.

Although this study is potentially interesting, there are problems that need to be addressed. These can be divided into major and minor concerns.

(A) MAJOR:

1. Copper-free click chemistry has been well described by previous investigators and has been applied for site-selective labeling of biomolecules in vivo (references 8-16). Although this is the first study to apply the technology to drug reversal, the novelty is somewhat limited.
2. Important controls are missing. The authors should (a) show that in contrast to its effect on WN3, BCN fails to reverse the anticoagulant effect of warfarin to highlight the specificity of BCN for an azide-modified target, and (b) compare the capacity of BCN to reverse the anticoagulant effect of WN3 with that of vitamin K, the current antidote for warfarin. The latter study is critical to determine whether BCN offers advantages over vitamin K for rapid WN3 reversal.
3. Warfarin exerts its anticoagulant effect by reducing the synthesis of the vitamin K-dependent clotting proteins, factors II, VII, IX and X. Its onset of action is delayed because it takes 4-5 days to lower the levels of prothrombin into the therapeutic range. Prolongation of the PT by warfarin reflects the reduction in the levels of factors II, X and VII. Of these factors, factor VII has the shortest half-life. With this background, I have concerns with the studies described in this paper. First, it is questionable whether any of the studies were performed when WN3 was at steady state. Mice were given only 2 doses of the drug 24 hours apart. Therefore, the rapid prolongation of the PT may have reflected a decrease in factor VII levels without a concomitant decrease in the levels of factors X and II. To address this concern, the authors should measure the levels of factors II, VII and X before and after WN3 administration. Second, BCN was given within minutes of the second dose of WN3. This is not the scenario that is likely to occur in patients nor does it make pharmacological sense because peak WN3 levels are not achieved until at least 10 min after dosing (Figure 5a). Why would the reversal agent be given prophylactically? Third, BCN lowers the PT within 6 h of administration. I can see how it can bind WN3 but how does it promote such rapid recovery in the clotting factor levels? Is this a factor VII-directed effect because factor VII has a half-life of 4-6 h or is it affecting the levels of the other vitamin K-dependent factors, which have larger half-lives. Measurement of clotting factor levels and head-to-head comparison of BCN with vitamin K will help to address these concerns.
4. Warfarin is highly protein-bound. Does WN3 bind albumin to the same extent as warfarin? Can BCN neutralize protein-bound WN3 as rapidly as free WN3?
5. Complete neutralization of the anticoagulant effect of WN3 by BCN requires a large molar excess of BCN over WN3. What is the explanation for this phenomenon? Does it reflect the binding of BCN and/or WN3 to plasma proteins? Given the need for a large molar excess of BCN over WN3, how would the dose of BCN be selected in practice when the concentration of WN3 is unknown? Is there a dose-response relationship between WN3 plasma concentrations and PT prolongation? Could this information be exploited for BCN dose selection?

(B) MINOR:

1. Page 2: Development of the aptamer-anti-aptamer pair described in references 1 and 3-5 has

been halted.

2. Page 2: The concept of "instant drug deactivation" is unlikely to apply to drugs like warfarin because even if the warfarin derivative is neutralized, it takes time to restore the synthesis of the vitamin K-dependent clotting factors.

3. Page 3: How do the kinetics of vitamin K epoxide reductase inhibition by WN3 compare with those of warfarin?

4. Page 5: The time dependence of the WN3 effect on the PT (Figure 3) is consistent with progressive lowering of the vitamin K-dependent clotting factors. The authors should acknowledge this point.

5. Page 6: BCN was administered subcutaneously. This is not an ideal route of administration for an antidote in a patient with a coagulopathy because (a) it can induce bleeding, and (b) the subcutaneous tissue may be poorly perfused in patients requiring warfarin reversal. Can BCN be given intravenously? Does it work faster when given this way?

6. Page 9: The authors raise the possibility that this "click and clear" strategy would permit the development of drugs with suitable biophysical properties. Would it not make sense to have a universal reversal agent such as BCN that could neutralize any azide-labeled drug?

7. There are numerous grammatical errors that require attention.

Reviewer #1 (Remarks to the Author):

Wagner and colleagues describe an appealing approach to modulate the properties of a drug, this case warfarin, by means of click chemistry *in vivo*. To temporally control the extent of warfarin activity they modified it with an azide and were able to rapidly deactivate and clear the active drug by administering a cyclooctyne agent. As a result the anticoagulation activity was reduced ca 50 %. The work has a potential impact on the way certain long lasting drugs can be designed. The experimental approach and data presentation is good.

In my opinion this paper can become suitable for publication, provided below issues are adequately addressed.

In the introduction the authors state:

" While these results establish the feasibility of using *in vivo* bioorthogonal chemical reaction in living organism when one reactant is targeted to specific tissues (via a labeled antibody or a polymer), the possibility of neutralizing a circulating drug that requires bioorthogonal reactions to proceed at the level of the whole organism still remained unknown."

The authors have omitted the work from Rossin *et al* in *J. Nucl. Med.* 2013, 54, 11, 1989-1995. Clearing agents based on tetrazine-modified albumin or microparticles were used to react (click) with and rapidly clear freely circulating trans-cyclooctene-modified antibody from blood in mice.

Answer: We have missed this publication which is indeed of high interest. As suggested by the reviewer, we included this work in the introduction of our paper:

"...In pioneer work Rossin R. *et al.* developed a novel 2-tetrazine-functionalized clearing agents that enable the rapid reaction with and the removal of a TCO-tagged antibody from blood. The use of such clearing agent led to 125-fold improvement of the Ab tumor-to-blood ratio 3 h after tetrazine injection. This was achieved through rapid reaction of clearing agents with the mAb in blood to remove it from the circulation and concentrate it in the liver or spleen without blocking the tumor-bound mAb."
."

In addition to including this work in the introduction, the paper will be stronger if the authors compare their click&clear design and corresponding results to the other published click&clear design.

To summarize the difference between our click and clear strategy and the published drug neutralizing approach we have rephrased part of the introduction:

"... Another approach focuses on the development of a novel type of aptamer-based anticoagulant that can be neutralized with antisense oligonucleotides, porphyrines, or positively charged oligomers. The system REG 1 comprising an aptamer as reversible antagonists of coagulation factor IXa and its complementary oligonucleotides as antidote had been developed until clinic trials and, to the best of our knowledge, has been halted in phase III. Both of these approaches rely on specific non-covalent interaction to scavenge the exogenous compound preventing its biological effect without however removing it from the organism.

To address the issue of drug neutralization and clearance, we have investigated the opportunity offered by *in vivo* bioorthogonal chemistry. Our idea was to design a small-molecule drug containing an azide group that would act as a safety pin, allowing drug deactivation and clearance via *in vivo* click reaction with a suitable strained cyclooctyne-containing neutralizing agent ..."

The authors observed a ca 10-fold increased reactivity in serum compared to PBS and suggest that this could be due to hydrophobic interactions between BCN and serum proteins. Usually these click reactions are accelerated when going from organic solvents to water, which does not seem to fit with above explanation. In addition such interactions surely would decrease the availability of the BCN for the azide?

We had recorded kinetic of reaction between WN_3 and unmodified BCN in both PBS and plasma. We have found kinetic constant of 0.8 and $11.1 \text{ M}^{-1}\text{s}^{-1}$ in respectively PBS and plasma. These data were presented in SI (Figure S2).

We totally agreed with comment of the reviewer that click reactions are accelerated when going from organic solvents to water. This is actually the reason why we underlined this unusual observation. These kinetic experiments have been repeated now three times and similar results were obtained.

Additional kinetic experiments using BCN-peg₆-OH (9) have been added. First kinetic constant of reaction between WN_3 (2) and BCN-peg₆-OH (9) was found to be $6.3 \text{ M}^{-1}\text{s}^{-1}$ in the plasma (see SI Figure S6) which is of the same order of magnitude as the result obtained with non-modified BCN ($11.1 \text{ M}^{-1}\text{s}^{-1}$). Then, to check the availability of WN_3 (2) and BCN-peg₆-OH (9) in the plasma for the click reaction, we performed kinetic experiments after pre-incubation of WN_3 (2) or BCN-peg₆-OH (9) in the plasma at 37°C. WN_3 (2) pre-incubation in the plasma didn't affect its reactivity toward BCN-peg₆-OH (9) (SI Figure S8a).

Concerning BCN-peg₆-OH (9), it is indeed known that its stability is limited in plasma. To evaluate the effect of this degradation on its ability to neutralize WN_3 (2) in plasma we have recorded kinetic after pre-incubation of BCN-peg₆-OH (9) for 0, 2h30 and 15h in plasma (SI Figure S8b-c). These results showed that stability of BCN in plasma was sufficient in regard of its time of residence in mice organism to engage efficient click and clear strategy.

To address this point in our manuscript we have added:

"...The reactivity of WN_3 2 was evaluated with both BCN and BCN-peg₆-OH 9. Kinetic experiments were recorded in PBS and in plasma at 100 μM with both reagents. Corresponding rate constants of 0.81 and $11.1 \text{ M}^{-1}\text{s}^{-1}$ were found for BCN (see SI Figure S2). By using 10 equivalents of BCN (1 mM), the conversion of WN_3 2 into clicked product 10 reached 90% in 15 min (see SI Figure S5). The kinetic constant for BCN-peg₆-OH 9 was found to be $6.3 \text{ M}^{-1}\text{s}^{-1}$ in the plasma (see SI Figure S6), which is of the same order of magnitude than non-modified BCN ($11.1 \text{ M}^{-1}\text{s}^{-1}$), while lower than in water."

We have also added in the section:

" WN_3 2 was found to be stable when kept at 37°C in the plasma or as dried powder for more than 24 h without particular precaution. WN_3 2 stability in plasma was also assessed by recording the kinetic of neutralization with BCN-peg₆-OH 9 after incubation of a WN_3 2 plasmatic solution at 37°C for 15h. In this condition a reaction rate constant of $4.8 \text{ M}^{-1}\text{s}^{-1}$ was measured (SI Figure S8a) similar to the control experiment ($6.3 \text{ M}^{-1}\text{s}^{-1}$, SI Figure S6).

BCN is known to be a reactive scaffold with limited stability in biological media. We thus decided to evaluate the effect of the time of residence of BCN-peg₆-OH 9 in the plasma at 37°C on its ability to neutralize WN_3 2. As expected, we found that kinetic constants recorded at 100 μM , 37°C in plasma, slowly decreased with time. Reaction rate constant was found to be 6.3, 2.8 and $1.5 \text{ M}^{-1}\text{s}^{-1}$ respectively after 0, 2h30 and 15h. As BCN-peg₆-OH 9 reaches its maximum plasmatic concentration at 20 min and is mainly cleared in the urine after 45 min (Figure 7), we considered that BCN-peg₆-OH 9 showed a suitable reactivity vs stability balance to use it as clearing agent for *in vivo* experiments."

As a side note, the authors should note the concentrations used in this experiment to accompany the statement regarding the complete reaction yield in 15 minutes.

This point is described in SI (4. Kinetic studies) but more clarity was added in the main text:

"...Kinetic experiments were recorded in PBS and in plasma at 100 μM with both reagents..."

The authors achieve effective deactivation/clearing of warfarin (Figure 5), but the effect on anticoagulation appears to be modest: ca 2 fold. Could this be due to re-equilibration of warfarin from other tissues into circulation after the warfarin the blood has cleared ? The studied time frame in Figure 5 is 1 h while the results in Figure 6 are after 6 hours, so possibly some re-equilibration could have occurred?

We agree with reviewer that limited anticoagulant effect might be related to either retention of WN_3 (2) in the liver or to its rapid metabolism leading to active metabolites thus reducing its availability for reaction with BCN in the plasma. To address this important point we have measured the amount of WN_3 (2) remaining in the liver after BCN-peg₆-OH (9) treatment. Accordingly, at 45 minutes after BCN-peg₆-OH (9) treatment there is almost no WN_3 (2) detectable in the liver (4.3 nmol/g) as compared to 31.4 nmol/g observed at the same time-point in mice injected with WN_3 (2) alone (see Figure 7).

Pharmacokinetic studies described in Figure 5 showed that WN_3 has a long circulation time. To answer concerns of reviewer 2, we have shown that WN_3 (2) is highly plasma protein-bound (98%, see SI 8. Plasma protein binding of Warfarin derivatives). This indicates that most of available warfarin remains in the bloodstream and is not accumulated in the various organs. This latest data is not in favour of re-equilibration from organs postulate.

Another explanation for this remaining anti-coagulant activity might arise from the fact that our neutralizing strained alkyne was designed to increase clearance of the WN_3 adduct. Indeed our new pharmacokinetic data show that amount of BCN-peg₆-OH (9) available for click reaction is limited by its elimination in the urine (1524 nmol/mL in urine vs 70 nmol/mL in plasma 45 min after BCN injection, Figure 7). These interesting results indicate that further improvement in click and clear efficiency lies in improving the balance between reactivity and excretion profile.

To support this, we add new data as Figure 7 and add the following sentence:

"However, the PT measures after neutralization of WN_3 2 by BCN-peg₆-OH 9 were still above the PT of control mice, which could be due to re-equilibration of WN_3 2 into circulation from other tissues such as the liver. To address this important point, we have measured the amount of WN_3 2 remaining in the liver after BCN-peg₆-OH 9 treatment. Accordingly, at 45 minutes after BCN-peg₆-OH 9 treatment, there are low but detectable levels of free WN_3 2 in the liver; 4.3 nmol/g as compared to 31.4 nmol/g observed at the same time-point in mice injected with WN_3 2 alone (compare "no click reaction" and "in vivo click reaction" panels in **Error! Reference source not found.**). Although such data suggest that re-equilibration of the WN_3 2 from the liver could account at least in part for incomplete normalization of PT, we cannot exclude an alternative possibility that neutralizing strained alkyne, which was designed to increase clearance of the WN_3 2 adduct, is in fact eliminated from circulation prior to click reaction. We thus measured the renal clearance of WN_3 2, BCN-peg₆-OH 9 and clicked product 10."

With respect to the previous point and in general, quite some background information is missing:

- what is known about warfarin PKPD? should it indeed re-equilibrate from other tissues back into the blood?

Previous studies have shown that warfarin binds extensively to blood plasma proteins and that only a small fraction of the drug is unbound and thus available for therapeutic function.¹ Warfarin is essentially completely absorbed, reaching a maximum plasma concentration between 2 and 6 hours. It distributes into a small volume of distribution (10 L/70kg) and is eliminated by hepatic metabolism with a very small clearance (0.2 L/h/70kg). The elimination half-life is about 35 hours. Most of available warfarin remains in the bloodstream and is not accumulated in the various organs which is not in favour of re-equilibration from organs postulate

- Going back to the previously quoted sentence in the introduction:

"the possibility of neutralizing a circulating drug that requires bioorthogonal reactions to proceed at the level of the whole organism still remained unknown."

Does this mean that the authors expect that their BCN agent extravasates and can deactivate warfarin reservoirs outside the vasculature?

We thank the reviewer for this wise comment. Indeed we do not claim to extravasate and deactivate warfarin reservoir outside the vasculature. This was a wording mistake and we corrected the sentence:

“...With this demonstration that click chemistry could proceed in the vasculature at the level of the whole organism, we decided to study the neutralization...”

However, to investigate this point we hypothesized that if WN_3 (2) accumulates in organs, where it could become inaccessible for BCN, the efficiency of the clearing agent injection would be decreasing proportionally to the delay between WN_3 (2) and BCN-peg₆-OH (9) administration. We recorded new data that does not support this possibility as BCN mediated PT reduction was similar for BCN injected at 5, 10 or 30 min after WN_3 (2) treatment (see SI Figure S11). These latest results indicate that indeed WN_3 (2) remained mostly in the bloodstream where BCN-peg₆-OH (9) can neutralize it.

- *What is the reason for the very slow clearance of warfarin and their azido-warfarin? Could it be due to interaction with albumin for example ?*

As above discussed previous studies have shown that warfarin binds extensively to blood plasma proteins and that only a small fraction of the drug is unbound and thus available for therapeutic function.

Our measurement of WN_3 (2) plasmatic protein binding is consistent with this previous report as shown in SI (SI 9. Plasmatic protein binding of warfarin derivatives).

- *Do the used warfarin concentrations correspond to clinical concentrations (when accounted for mouse vs man) ?*

The doses of WN_3 (2) used in our mouse experiments correspond to doses of Warfarin used in clinical conditions as warfarin dose used in human varies to attain 2-4 fold increase of PT, whereas using 2 mg/kg of WN_3 in mouse we attain about 3.5 fold increase of PT (see Figure 4). We have now added this information in the text :

“...Moreover, this dose of WN_3 2 corresponds to doses of Warfarin 1a used in clinical conditions as Warfarin dose used in human varies to attain 2-4 fold increase of PT...”

- *Is a 50 % reduction of anti-coagulation activity clinically relevant? What is the clinic looking for in this respect?*

Yes, according to medical doctors working at the “Etablissement Français du Sang”, reduction of anticoagulant activity by 50% is of direct medical interest. We would like, however, to insist on the fact that clinical application and implications of our study are beyond the scope of our manuscript in which we use WN_3 (2) treatment and its functional readout (parameters of coagulation) as a convenient tool to demonstrate the possibility of inactivating and clearing an active pharmacological compound in the organism of a living animal using click reaction.

- *How does the efficacy of their approach compare to those referenced in the introduction?*

Please also note that antidote approaches described in the introduction relies on non-covalent interaction between the targeted drug and the neutralizing agent to scavenge it. However with this strategy no clearance from the organism is possible.

We added in the introduction:

“...Both of these approaches rely on specific non-covalent interaction to scavenge the exogenous compound preventing its biological effect without however removing it from the organism.”

In addition, the first approach (cyclodextrine) presented in the introduction is hampered by the need to design complex molecular cages able to form precise supra-molecular interaction with the drug while keeping good biocompatibility. On the other hand our approach relies on the design of a neutralizing agent which should have low toxicity and biodistribution profile compatible with the

neutralization of the drug of interest. Both strategies seem quite complementary each of them bearing their own limitations.

The aptamer approach also relies on non-covalent interaction and do not result in clearance of the bioactive compound. It is moreover severely limited by the known poor aptamer biostability. To the best of our knowledge the only compound that still in clinic (macugen) is administered by local intra-ocular injection.

Minor comments:

- *the last line of the second paragraph in the introduction should provide one or more references.*

We added in the introduction two references concerning the original development of cyclodextrine as reversal agents for the neuromuscular blocker^{ii,iii} (References 1 and 2 of the manuscript) and also two reviews centered on the use of Sugammadex in clinic (References 3 and 4 of the manuscript).^{iv,v}

- *faulty grammar in the last line of paragraph 4 of the introduction*

This sentence has been modified:

“In this strategy, the new molecule formed by reaction of the drug and the antidote should be ideally deprived of any biological activity and quickly cleared from the bloodstream *via* a renal excretion”

- *BCN-oeg6-OH should be BCN-peg6-OH*

BCN-oeg₆-OH (9) has been replaced by BCN-peg₆-OH (9).

- *section "Click in vivo, metabolomic analysis": line 5 is unclear. Amongst others by confusing use of "addition". Perhaps better to use "the sum".*

For clarification purposes we modified the manuscript:

“...Plasma samples were collected at 10, 20, 30 and 60 min after the administration of BCN-peg₆-OH **9**. The clicked product **10** was detected at a concentration of 3.1 ± 0.6 nmol/mL already at 10 min, indicating that the reaction took place *in vivo* (**Error! Reference source not found.b**). Furthermore, inactivated drug **10** appeared to be the sole product arising from WN₃ **2** since **the sum** of the concentration of the remaining WN₃ **2** (1.0 ± 0.5 nmol/mL, **Error! Reference source not found.c**) and the clicked product **10** (3.1 ± 0.6 nmol/mL) corresponds to maximal plasma concentration of WN₃ **2** expected in absence of BCN-peg₆-OH **9** (4.2 ± 0.4 nmol/mL; compare **Error! Reference source not found.a**). Accordingly, the concentration of circulating BCN-peg₆-OH **9** was reduced (compare **Error! Reference source not found.b** and d). Most notably, these results also revealed that the click reaction between **2** and **9** seemed to trigger quick drug elimination because there was no more free WN₃ **2** nor click product **10** in plasma 60 min after administration of BCN-peg₆-OH **9** (**Error! Reference source not found.c**). This is consistent with the pharmacokinetic profile of BCN-peg₆-OH **9** for which plasmatic concentration diminished faster after reaching its peak of plasma concentration than does WN₃ **2** (**Error! Reference source not found.a** and b)...”

- *section "WN3 (2) inactivation by click in vivo": the term "PT fold change" is unclear and also faulty. Consider changing into for example "PT ratio"*

To facilitate comparisons between different experimental conditions and clinical data, changes of PT measures were reported as fold change with respect to PT values in control mice. We have now specified in the text what we mean by "fold change" (p.6 line 4 from the top):

“As a result of these analyses, we retained for further functional studies an intraperitoneal injection of 2 mg/kg and PT measurement after 6 hours as optimal experimental conditions to investigate *in vivo* drug inactivation, providing significant, about 3.5-fold increase of anticoagulant activity (as compared to the control group) with low variability.”

We also specified how such "fold change" was calculated in the Material and Methods section (the last sentence of the paragraph "Formulation of Warfarin and BCN derivatives and pharmacological treatments"):

"...To facilitate comparisons between different experimental conditions and clinical data, changes of PT measures were reported as fold change with respect to PT values in control mice, and were calculated as the mean of ratios of each individual PT value from a specific treatment group to the mean PT value of control group."

- section *Methods, Metabolomic analysis*: should "minimum 10 min" be "maximum 10 min"?

To reduce an error of plasma preparations due to very short periods of incubation and in consequence incomplete plasma preparation we left blood samples at room temperature for at least 10min prior to plasma separation (protocol used at Mouse Clinical Institute). This information was added in materials and methods section: LC-MS/MS analyses:

"...This minimal incubation period aimed to reduce an error which could occur at very short periods of incubation and in consequence incomplete plasma preparation."

- *Metabolomic should be Metabolic*

Metabolomic analyses have been replaced by LC-MS/MS analyses.

ⁱ Holford, N.H.G. Clinical pharmacokinetics and pharmacodynamics of Warfarin. *Clinical Pharmacokinetics* **11**, 483 (1986).

ⁱⁱ Bom, A., Bradley, M., Cameron, K., Clark, J. K., van Egmond, J., Feilden, H., MacLean, E. J., Muir, A. W., Palin, R., Rees, D. C. & Zhang, M.-Q. A novel concept of reversing neuromuscular block: chemical encapsulation of rocuronium bromide by a cyclodextrin-based synthetic host. *Angew. Chem. Int. Ed.* **41**, 266 (2002).

ⁱⁱⁱ Adam, J. M., Bennett, D. J., Bom, A., Clark, J. K., Feilden, H., Hutchinson, E. J., Palin, R., Prosser, A., Rees, D. C., Rosair, G. M., Stevenson, D., Tarver, G. J. & Zhang, M.-Q. Cyclodextrin-derived host molecules as reversal agents for the neuromuscular blocker rocuronium bromide: synthesis and structure–activity relationships. *J. Med. Chem.* **45**, 1806 (2002).

^{iv} Kovac, A. L., Sugammadex: the first selective binding reversal agent for neuromuscular block. *J Clin. Anesth.* **21**, 444 (2009).

^v Yang, L. P., Keam, S. J. Sugammadex: a review of its use in anaesthetic practice. *Drugs* **69**, 919 (2009).

Reviewer #2 (Remarks to the Author):

Wagner and colleagues describe a "click & clear" approach to control the anticoagulant activity of warfarin analog. The authors synthesized an azido warfarin derivative and characterized its activity *in vivo*. The azido drug could be covalently targeted with a complementary cyclooctyne; the "clicked" conjugate harbored no anticoagulant properties. Thus, the bioorthogonal ligation could effectively turn off warfarin activity. While this is an interesting concept, I have some significant concerns about the experiments and overall approach (see below). I do not recommend publication at this time.

1. The mechanism of anti-coagulant activity *in vivo* is still a bit nebulous. The authors demonstrate that the cyclooctyne alone can impact anti-coagulant activity (Figure 4). The data in Figure 6 are incredibly difficult to interpret. How is it possible to rule out that the cyclooctyne itself (and not the "clicked" conjugate) is responsible for the PT differences? Perhaps more rigorous competition experiments are necessary.

In fact, PT observed in wild-type animals treated with BCN alone were not significantly lower from vehicle treated mice and p value was now reported in text (line 9 in the paragraph "Anticoagulant activity of clearing cycloalkyne (9) and clicked product (10)").

However, we do agree with reviewer that such controls are not sufficient as they can be affected by "floor effect" of PT measures and that one important control was missing to exclude the possibility that BCN-peg₆-OH (9) on its own may have pro-coagulant activity in conditions when PT is high. We have now tested activity of BCN-peg₆-OH (9) after administration of warfarin 1a (and not WN₃ 2) which was not modified with an azide group and thus which should be not sensitive to click reaction with BCN-peg₆-OH (9). We did not observe any reduction of PT in mice injected with warfarin and BCN-peg₆-OH 9 indicating clearly that BCN-peg₆-OH (9) on its own does not display any pro-coagulant activity. This additional data is shown in Figure S10 (SI).

In addition we added in the main text:

"...Since all measures of BCN-peg₆-OH 9 activity were performed in non-stimulated conditions when PT values are naturally low, we could not exclude the possibility that some endogenous pro-coagulant activity of BCN-peg₆-OH 9 would be masked by the "floor effect" of PT measurements. To address this point, we tested activity of BCN-peg₆-OH 9 in mice in which PT was increased by treatment with Warfarin 1a (which does not bear an azide group) and thus which should be not sensitive to click reaction. As expected, using the same protocol of injections as for WN₃ 2 we did not observe any reduction of PT in mice injected with 2 mg/kg of Warfarin 1a followed by 20eq of BCN-peg₆-OH 9, indicating clearly that BCN-peg₆-OH 9 on its own does not display any pro-coagulant activity (Figure S10). In line with this observation, BCN-peg₆-OH 9 did not display any major effect on Warfarin 1a mediated reduction of clotting factors (Figure S12)..."

To make the reading of the manuscript easier, we decided to concentrate the discussion around BCN-peg₆-OH (9) activity in Figure 4 and in the comment below it and to discard the redundant information in Figure 6.

2. The authors do not make a strong case for the "clickable" antidote. One could argue that the effects of warfarin could be more easily tuned using other small molecule inhibitors, etc. that target other relevant pathways. The "click" reaction could be an unnecessary complication. (Note that large concentrations of these reagents are necessary for efficient *in vivo* ligations.)

To make a stronger case of the clickable antidote strategy, we included a clearer description how the efficacy of this approach compared to those already described. We added in the introduction:

"...Both of these approaches rely on specific non-covalent interaction to scavenge the exogenous compound preventing its biological effect without however removing it from the organism.

To address the issue of drug neutralization and clearance, we have investigated the opportunity offered by *in vivo* bioorthogonal chemistry. Our idea was to design a small-molecule drug containing an azide group that would act as a safety pin, allowing drug deactivation and clearance *via in vivo* click reaction with a suitable strained cyclooctyne-containing neutralizing agent. In this strategy, the new molecule formed by reaction of the drug and the antidote should be ideally deprived of any biological activity and quickly cleared from the bloodstream *via* a renal excretion..."

Please also note that antidote approaches described in the introduction relies on non-covalent interactions between the targeted drug and the neutralizing agent to scavenge it. With this strategy no clearance from the organism was showed. To illustrate the advantage of the click and clear strategy, we have performed additional experiments to demonstrate that our approach result not only in the inactivation of the drug but also leads to fast renal clearance. We measured the renal clearance of WN₃ (2), BCN-peg₆-OH (9) and clicked product (10) (Figure 7) and showed that the clearing agent (BCN-peg₆-OH) and the clicked product 10 were mainly eliminated via the urine 45 min after BCN-peg₆-OH 9 injection.

In our paper, we used WN₃ (2) and its reaction with BCN as a paradigm treatment, which allows taking advantage of well-established techniques to control molecular and functional readout of WN₃ activities on blood coagulation. We therefore used this pharmacological treatment as a model to demonstrate the possibility of inactivating an active pharmacological compound in the organism of a living animal using click reaction.

The possibility of targeting WN₃ in the living organism and inactivating it by click reaction together with our new data on the rapid elimination of clicked product (Figure 7) and efficiency of BCN-peg₆-OH (9) activity at different time-points after WN₃ injection (see SI Figure S11) do support the idea that our approach could be used to develop clickable antidote in blood coagulation management.

3. I had a difficult time assessing the quality of some experiments based on the few details provided. The PT assay itself was not well described. Sample data/spectra for key LCMS studies were also not provided. Such studies are notoriously difficult to perform in the absence of radiolabels, etc. Additionally, no NMR spectra were provided to assess purity.

Experimental data and new experimental methods have been added in the supporting information:

-¹H NMR, ¹³C NMR of WN₃ (2), BCN-peg₆-OH (9) and clicked product (10)

-HPLC monitoring of SPAAC kinetic studies

-Sample preparation description (blood, urine and liver) for LC-MS/MS analyses

-LC-MS/MS calibration in plasma, urine and liver

-Procedure description of PT and coagulation factors evaluation

-Procedure for plasmatic protein binding studies and LC-MS/MS analyses

4. Aryl azides and cyclooctynes are not completely stable/inocuous. (Aryl azides are also difficult to characterize via conventional HPLC/LCMS owing to photolysis issues; thus, the lack of data noted in point #3 above are all the more important to include.) Cyclooctynes are also known to be sequestered by numerous serum proteins and membranes. Did the authors observe such phenomena? (Such studies might require radiolabels or some other identifiable tag). Off-target labeling could be detrimental to in vivo systems.

We checked the stability of WN₃ (2) by leaving it in plasma, PBS and dry powder at room temperature over the bench without particular precautions and did not observe any degradation. We furthermore added kinetic experiments with WN₃ pre-incubated in plasma for one day at 37°C (SI Figure S8a).

Concerning BCN-peg₆-OH (9), it is indeed known that its stability is limited in plasma. To evaluate the effect of this degradation on its ability to neutralize WN₃ (2) in plasma we have recorded kinetic after pre-incubation of BCN-peg₆-OH (9) for 0, 2h30 and 15h in plasma (SI Figure S8b-c). These results showed that stability of BCN in plasma was sufficient in regard of its time of residence in mice organism to engage efficient click and clear strategy.

We added these data in SI as Figure S8 and added in the main text:

“WN₃ (2) and BCN-peg₆-OH (9) plasmatic stability. WN₃ 2 was found to be stable when kept at 37°C in the plasma or as dried powder for more than 24 h without particular precaution. WN₃ 2 stability in plasma was also assessed by recording the kinetic of neutralization with BCN-peg₆-OH 9 after incubation of a

WN₃ **2** plasmatic solution at 37°C for 15h. In this condition a reaction rate constant of 4.8 M⁻¹s⁻¹ was measured (SI Figure S8a) similar to the control experiment (6.3 M⁻¹s⁻¹, SI Figure S6). BCN is known to be a reactive scaffold with limited stability in biological media. We thus decided to evaluate the effect of the time of residence of BCN-peg₆-OH **9** in the plasma at 37°C on its ability to neutralize WN₃ **2**. As expected, we found that kinetic constants recorded at 100 μM, 37°C in plasma, slowly decreased with time. Reaction rate constant was found to be 6.3, 2.8 and 1.5 M⁻¹s⁻¹ respectively after 0, 2h30 and 15h. As BCN-peg₆-OH **9** reaches its maximum plasmatic concentration at 20 min and is mainly cleared in the urine after 45 min (Figure 7), we considered that BCN-peg₆-OH **9** showed a suitable reactivity vs stability balance to use it as clearing agent for *in vivo* experiments.”

Reviewer #3 (Remarks to the Author):

To evaluate the potential clinical utility of copper-free click chemistry, Ursuegui and colleagues synthesized an azido-modified derivative of warfarin (designated WN3) and examined the capacity of a cyclooctyne reagent to neutralize its anticoagulant activity in mice. The authors show that (a) WN3 prolongs the prothrombin time (PT) in a time- and dose-dependent manner, and (b) cyclooctyne reagent, which they designate BCN-peg6-OH (BCN) reverses the PT prolongation produced by WN3. Based on these findings, they conclude that this "click and clear" strategy provides a framework for developing reversible drugs.

Although this study is potentially interesting, there are problems that need to be addressed. These can be divided into major and minor concerns.

(A) MAJOR:

1. Copper-free click chemistry has been well described by previous investigators and has been applied for site-selective labeling of biomolecules *in vivo* (references 8-16). Although this is the first study to apply the technology to drug reversal, the novelty is somewhat limited.

To make a stronger case of the clickable antidote strategy, we included a clearer description how the efficacy of this approach compared to those already described. We added in the introduction:

"...Both of these approaches rely on specific non-covalent interaction to scavenge the exogenous compound preventing its biological effect without however removing it from the organism.

To address the issue of drug neutralization and clearance, we have investigated the opportunity offered by *in vivo* bioorthogonal chemistry. Our idea was to design a small-molecule drug containing an azide group that would act as a safety pin, allowing drug deactivation and clearance *via in vivo* click reaction with a suitable strained cyclooctyne-containing neutralizing agent. In this strategy, the new molecule formed by reaction of the drug and the antidote should be ideally deprived of any biological activity and quickly cleared from the bloodstream *via* a renal excretion."

Please also note that antidote approaches described in the introduction relies on non-covalent interactions between the targeted drug and the neutralizing agent to scavenge it. With this strategy no clearance from the organism was showed. To illustrate the advantage of the click and clear strategy, we have performed additional experiments to demonstrate that our approach result not only in the inactivation of the drug but also leads to fast renal clearance. We measured the renal clearance of WN₃ (2), BCN-peg₆-OH (9) and clicked product (10) (Figure 7) and showed that the clearing agent (BCN-peg₆-OH) and the clicked product were mainly eliminated *via* the urine 45 min after BCN-peg₆-OH (9) injection.

In our paper, we used WN₃ (2) and its reaction with BCN as a paradigm treatment, which allows taking advantage of well-established techniques to control molecular and functional readout of WN₃ activities on blood coagulation. We therefore used this pharmacological treatment as a model to demonstrate the possibility of inactivating an active pharmacological compound in the organism of a living animal using click reaction.

The possibility of targeting WN₃ in the living organism and inactivating it by click reaction together with our new data on the rapid elimination of clicked product (Figure 7) and efficiency of BCN-peg₆-OH (9) activity at different time-points after WN₃ injection (see SI Figure S11) do support the idea that our approach could be used to develop clickable antidote in blood coagulation management.

2. Important controls are missing. The authors should (a) show that in contrast to its effect on WN3, BCN fails to reverse the anticoagulant effect of warfarin to highlight the specificity of BCN for an azide-modified target, and (b) compare the capacity of BCN to reverse the anticoagulant effect of WN3 with that of vitamin K, the current antidote for warfarin. The latter study is critical to determine whether BCN offers advantages over vitamin K for rapid WN3 reversal.

We have now performed suggested experiments which indicate that BCN-peg₆-OH (9) displays anticoagulant effect only if applied in WN₃ treated mice whereas its administration in mice treated with non-modified warfarin (W) did not reduce PT measures (SI Figure S10). We added in the main text:

“To address this point, we tested activity of BCN-peg₆-OH **9** in mice in which PT was increased by treatment with Warfarin **1a** (which does not bear an azide group) and thus which should be not sensitive to click reaction. As expected, using the same protocol of injections as for WN₃ **2** we did not observe any reduction of PT in mice injected with 2 mg/kg of Warfarin **1a** followed by 20eq of BCN-peg₆-OH **9**, indicating clearly that BCN-peg₆-OH **9** on its own does not display any pro-coagulant activity (Figure S10). In line with this observation, BCN-peg₆-OH **9** did not display any major effect on Warfarin **1a** mediated reduction of clotting factors (Figure S12).”

Reduction of PT obtained in WN₃ treated mice after BCN-peg₆-OH (9) application was similar to PT reduction obtained by using a high dose (20 mg/kg) of vitamin K1 (see SI Figure S11). We added in the text:

“The pro-coagulant effects of BCN-peg₆-OH **9** were comparable to activity of high dose (20 mg/kg) of vitamin K1 in WN₃ **2** treated animals with respect to both reduction of PT values (Figure S11) and decreased activity of all three clotting factors, II, VII and X (Figure S13).”

Importantly, we also showed that use of BCN-peg₆-OH (9) is associated not only with rapid inactivation of WN₃ (2), but also its rapid clearance from the organism in the urine (Figure 7). Such clearance provides a clear advantage over the current antidote for warfarin such as vitamin K which can only increase coagulation without affecting bioavailability of warfarin and needs to be administrated at high doses adjusted to each patient due to its potential toxicity.

3. Warfarin exerts its anticoagulant effect by reducing the synthesis of the vitamin K-dependent clotting proteins, factors II, VII, IX and X. Its onset of action is delayed because it takes 4-5 days to lower the levels of prothrombin into the therapeutic range. Prolongation of the PT by warfarin reflects the reduction in the levels of factors II, X and VII. Of these factors, factor VII has the shortest half-life. With this background, I have concerns with the studies described in this paper. First, it is questionable whether any of the studies were performed when WN3 was at steady state. Mice were given only 2 doses of the drug 24 hours apart. Therefore, the rapid prolongation of the PT may have reflected a decrease in factor VII levels without a concomitant decrease in the levels of factors X and II. To address this concern, the authors should measure the levels of factors II, VII and X before and after WN3 administration.

As suggested by reviewer 3 we have now performed such analyses. Obtained data indicate that 2 days of Warfarin (1a) or WN₃ (2) treatment (2 mg/kg, IP injection) is sufficient to induce a significant and concomitant decrease in the three vitamin K-dependent factors: II, VII and X synthesis (SI Figure S12-13).

We added in the text:

“...We also confirmed that the prolongation of PT after WN₃ **2** or Warfarin **1a** treatment results in the concomitant decrease of the three major vitamin K-dependent clotting factors, II, VII and X as revealed by PT measures using dedicated clotting factor assays (SI Figure S12-13).”

...Second, BCN was given within minutes of the second dose of WN3. This is not the scenario that is likely to occur in patients nor does it make pharmacological sense because peak WN3 levels are not achieved until at least 10 min after dosing (Figure 5a). Why would the reversal agent be given prophylactically?

We selected 5 min delay between BCN-peg₆-OH (9) and WN₃ (2) administration as it is directly relevant to the main objective of our study in which we aim to show that click reaction may take place in a living mouse leading to inactivation and clearance of the circulating drug. Thus, according to our pharmacokinetic analyses, a SC injection of BCN at 5 min after IP WN₃ (2) application (through a distinct route of administration) should lead to its maximal concentrations

in the blood at 10 and 20 min when WN₃ (2) attained already its maximal and plateau concentration in the blood.

We agree, however, with reviewer that BCN-peg₆-OH (9) applications at longer delays would be more informative about clinical potential of our approach, but also about the mechanism of WN₃ (2) in the organism (the point raised by other reviewers).

We addressed this issue and our new data indicate that BCN-peg₆-OH (9) administration delayed by 10 and 30 min can also inactivate WN₃ (2) with equal efficiency as BCN-peg₆-OH (9) administration at 5 min post-WN₃ (2) treatment (SI Figure S11). We added in the text:

“BCN-peg₆-OH 9 administration delayed by 10 and 30 min can also inactivate WN₃ 2 with equal efficiency as BCN-peg₆-OH 9 administration at 5 min post-WN₃ 2 treatment (SI Figure S11). This observation is important as BCN applied at 30 min should attain its maximal levels in the organism 10-20 min later, which corresponds to 40-50min after WN₃ 2 administration. We found that at this time point WN₃ 2 is present not only in plasma but also in target organs such as the liver (**Error! Reference source not found.**), suggesting that BCN-peg₆-OH 9 may react with WN₃ 2 and clear it not only from circulation, but also from target organs.”

Third, BCN lowers the PT within 6 h of administration. I can see how it can bind WN3 but how does it promote such rapid recovery in the clotting factor levels? Is this a factor VII-directed effect because factor VII has a half-life of 4-6 h or is it affecting the levels of the other vitamin K-dependent factors, which have larger half-lives. Measurement of clotting factor levels and head-to-head comparison of BCN with vitamin K will help to address these concerns.

Vitamin K-dependent factors synthesis was also evaluated after *in vivo* click reaction and after vitamin K1 treatment as positive control. In the both case antidote administration significantly increased the three clotting factors II, VII and X synthesis in WN₃ treated mice (SI Figure S13c-d), which may reflect not only the rapid inactivation of WN₃ (2), but also its rapid clearance from the circulation and liver, which was discussed above and shown in Figure 7.

4. Warfarin is highly protein-bound. Does WN3 bind albumin to the same extent as warfarin? Can BCN neutralize protein-bound WN3 as rapidly as free WN3?

Pharmacokinetic studies described in Figure 5 showed that WN₃ has a long circulation time. We have now shown that WN₃ (2) is highly plasmatic proteins-bound as well as parent warfarin, 97% and 94% respectively (see SI Figure S17-18). Kinetic measurement recorded with WN₃ (2) dissolved in plasma showed indeed that neutralization proceeds with high speed (constant rate of 6.3 M⁻¹s⁻¹, Figure S6). Thus BCN-peg₆-OH (9) is able to neutralize WN₃ (2) which is mainly bound to plasmatic proteins.

5. Complete neutralization of the anticoagulant effect of WN3 by BCN requires a large molar excess of BCN over WN3. What is the explanation for this phenomenon? Does it reflect the binding of BCN and/or WN3 to plasma proteins? Given the need for a large molar excess of BCN over WN3, how would the dose of BCN be selected in practice when the concentration of WN3 is unknown? Is there a dose-response relationship between WN3 plasma concentrations and PT prolongation? Could this information be exploited for BCN dose selection?

Our new data indicate that BCN-peg₆-OH is rapidly secreted in the urine, which is interesting in terms for safety of its use, but which strongly impede its availability to attain WN₃ (2). In fact, an incomplete inactivation of WN₃ (2) reflects most probably short time of residence of BCN-peg₆-OH (9) in the plasma which is strongly supported by PK and new excretion data (Figure 7).

Considering the PK profile (high plasmatic concentration reached rapidly within 10 min followed by fast renal excretion) of BCN-peg₆-OH (9), we believe that one SC bolus injection at high dose should be sufficient to neutralize and clear circulating WN₃ (2) whatever its concentration.

Moreover in Figure 3 we have shown that there is a dose-response between WN_3 (2) administered dose and PT prolongation at 1 and 2 mg/kg. In our experiment that at the highest dose of 2 mg/kg, 20 equivalents of BCN-peg₆-OH (9), were able to neutralize the circulating drug. We thus believe that this dose can be applied for any lower dose of WN_3 (2).

(B) MINOR:

1. Page 2: Development of the aptamer-anti-aptamer pair described in references 1 and 3-5 has been halted.

This information has been added in the introduction:

“The system REG 1 comprising an aptamer as reversible antagonists of coagulation factor IXa and its complementary oligonucleotides as antidote had been developed until clinic trials and, to the best of our knowledge, has been halted in phase III.”

2. Page 2: The concept of "instant drug deactivation" is unlikely to apply to drugs like warfarin because even if the warfarin derivative is neutralized, it takes time to restore the synthesis of the vitamin K-dependent clotting factors.

This was a wording mistake and the term “instant” was removed.

3. Page 3: How do the kinetics of vitamin K epoxide reductase inhibition by WN_3 compare with those of warfarin?

We did not address this question as we believe it is beyond the scope of this study that mostly aims at demonstrating the click and clear concept.

4. Page 5: The time dependence of the WN_3 effect on the PT (Figure 3) is consistent with progressive lowering of the vitamin K-dependent clotting factors. The authors should acknowledge this point.

As discussed above, the synthesis of vitamin K-dependent clotting factors has been evaluated after WN_3 (2) treatment and *in vivo* click reaction. We confirmed that as Warfarin, WN_3 (2) treatment lowered coagulation factors II, VII and X (Figure S11-12). We showed also that administration of BCN-peg₆-OH (9) or vitamin K1 increased strongly the synthesis of the three vitamin K1-dependent clotting factors.

5. Page 6: BCN was administered subcutaneously. This is not an ideal route of administration for an antidote in a patient with a coagulopathy because (a) it can induce bleeding, and (b) the subcutaneous tissue may be poorly perfused in patients requiring warfarin reversal. Can BCN be given intravenously? Does it work faster when given this way?

We have also performed IV injection of BCN-peg₆-OH (9) but did not include them in the paper because of high variability even though we had the same PT reduction.

6. Page 9: The authors raise the possibility that this "click and clear" strategy would permit the development of drugs with suitable biophysical properties. Would it not make sense to have a universal reversal agent such as BCN that could neutralize any azide-labeled drug?

Of course this will make sense. We acknowledge this point by adding in the conclusion the following sentence:

“This “Click & Clear” strategy thus opens interesting prospects not only for the development of new drugs with switchable biophysical properties, but also in the design of a universal reversal agent with adequate pharmacokinetic/biodistribution properties that could neutralize any azide-labeled drug.”

7. *There are numerous grammatical errors that require attention.*

We carefully proofread our manuscript to correct those mistakes and rephrase some sentences to clarify their meaning.

Reviewers' comments:

Reviewer #1 (Remarks to the Author):

I am still in favor of publishing this interesting work, but there is still one issue that really needs to be addressed. As it stands now I am afraid that the authors are making an error, which, if I am correct, I am sure they would not want to publish as such.

The 10-fold higher reactivity in plasma compared to PBS is very strange. The authors have repeated the experiments and say they have come to the same numbers. First of all, reaction kinetics should always be based on at least $n=3$ and should be reported with the error. This is still not the case in the present revision and in my opinion should be addressed. The more so because the authors use the reaction kinetics as a read out of compound stability/availability over time.

The reaction kinetics in plasma was measured by measuring the decrease of free (non-protein bound) WN3 in time, using normal warfarin as internal standard. However, as they show in SI section 9 both warfarin and WN3 almost completely bind to plasma proteins within 4 hours. How can one determine the reactivity in plasma if ca 95-98 % of the compound is not taken into account due to the fact that it binds to protein and is removed before analysis. Apparently the authors assume that WN3 and warfarin have exactly the same binding kinetics to plasma protein. However, there is no basis for this.

The experiment in SI section 9 only looks at one time point and already gives a slightly different result for both compounds. The work in section 9 should be expanded to earlier time points to understand how fast and to what extent both compounds bind to proteins. In addition there the effect of concentration on plasma binding kinetics will need to be understood to know if the reaction kinetics in plasma can be trusted. Even a slight difference in binding, also at the earlier time points will result in a different k_2 . This in my opinion is the reason for the 10-fold higher k_2 in plasma compared to PBS. While I guess one can still use the current plasma k_2 to make a statement about stability in time (of the non bound fraction), it cannot be used as an argument that WN3 is equally reactive and available in plasma and PBS.

The authors attempt to address this by preincubating WN3 in plasma for 15 hours followed by assessing the reaction kinetics. The authors state that the found k_2 of 4.8 is similar to the earlier found 6.3 (without preincubation), but without error bars this statement cannot be made. It is still a ca 25 % drop and in addition the 15 hour time frame does not match the binding kinetics to protein, as in section 9 of the SI the authors show that within 4 hours ca 98 % of the WF3 has bound to plasma protein.

As it stands now the paper only reports about the 2 % of WN3 that is not bound to protein, i.e. the soluble fractions. The protein bound fractions are removed before analysis; this also holds for the LCMS analysis of blood and liver samples. There is no evidence that protein bound WN3 reacts to BCN and that the product bound to protein is cleared. This should be made very clear to the reader. And as it stands now the k_2 in plasma cannot be presented as a k_2 in plasma.

Regarding my earlier comment that the effect of the complete clearance of WN3 on anticoagulation appears to be modest (ca 2 fold):

The issues listed above seem to give ample room for the possibility that protein bound WN3 acts a reservoir and re-equilibrates the WN3 levels in blood after the click and clear. Based on Figure 5 C this does not seem to happen in the 1st hour, as the WN3 levels are 0. However, the authors' statement (lines 372-375) that the modest effect on anticoagulation is the result of incomplete reaction of BCN with WN3 is incorrect based on these results in Figure 5C, as the WN3 clearance is complete. Therefore it is perhaps more likely that re-equilibration at later time points, between 1 and 5 h are the cause for the modest anticoagulation effect. At least this should be clearly addressed.

minor comments

line 137-8 " which is of the same order of magnitude than non-modified BCN (11.1 M-1s-1), while lower than in water."

"lower" should be "higher" ?

Figure 7 caption, please add the time

The grammar and spelling still needs attention , also in the SI

Reviewer #2 (Remarks to the Author):

I was a previous reviewer of this manuscript. The revised submission is substantially improved. Many of my concerns have been addressed. (I am not qualified to comment on whether the other reviewer concerns were addressed.)

I still have some issues with the following:

1. How the rate/kinetic data are being interpreted. The authors note that the "click" reaction rate increases in plasma. (This is counterintuitive based on the mechanism of the reaction and what other groups have reported.) From what I can tell in the SI, the authors are using the disappearance of starting materials to measure rate constants. This is decidedly less desirable than monitoring product production. (There are many ways that the starting materials can be "consumed", including sequestration on hydrophobic proteins, etc. So, the rate constants being reported do not accurately reflect the ligation rates in the environments. The authors' claims on rapid reactivity (and even the amounts of reagents necessary for in vivo ligation) could be skewed by over-interpreted data.

2. The presentation of reagent stability/robustness data. It is still not easy to see what the authors are claiming. Note in Figure S8 that no comparison data are provided to the non-heated/non-incubated controls. (Stability data are usually provided as half-life values or as "percent remaining"/normalized plots.)

Reviewer #3 (Remarks to the Author):

The authors have addressed my major concerns. However, there are some minor points that need to be addressed to improve the flow of information.

1. For consistency, the authors should either show the prothrombin time ratios (as they do in Figure 4) or the prothrombin times (as they do in Figures 3 and 4), but not a mixture of both.

2. The authors show the prothrombin times in factor II, VII or X-deficient plasma before and after warfarin or WN3 administration (Figure S12) and after WN3 reversal with vitamin K or BCN-peg6-OH (Figure S13). By comparison with a standard curve constructed by adding varying amounts of normal plasma, they need to use this information to determine the level of each of these clotting factors.

3. There still are numerous typographical and grammatical errors that require attention.

Reviewers' comments:

Reviewer #1 (Remarks to the Author):

I am still in favor of publishing this interesting work, but there is still one issue that really needs to be addressed. As it stands now I am afraid that the authors are making an error, which, if I am correct, I am sure they would not want to publish as such.

The 10-fold higher reactivity in plasma compared to PBS is very strange. The authors have repeated the experiments and say they have come to the same numbers. First of all, reaction kinetics should always be based on at least $n=3$ and should be reported with the error. This is still not the case in the present revision and in my opinion should be addressed. The more so because the authors use the reaction kinetics as a read out of compound stability/availability over time.

Answer: We acknowledge reviewer's warning and consequently performed additional kinetic experiments to address her/his concerns. We decided also to remove kinetic data concerning non-modified BCN from the manuscript. Indeed, the acceleration in plasma was observed with this model compound which is not the one used for in vivo experiments. Thus we now show in the article only kinetic data obtained with BCN-peg₆-OH **9**, the strained alkyne used for click and clear experiments. Nevertheless, kinetic experiments using non-modified BCN have been repeated both in PBS and plasma for reviewer only and results are discussed below. All the kinetic experiments have now been repeated three times and reported with standard deviation. Results remained in accordance with previously reported data.

The reaction kinetics in plasma was measured by measuring the decrease of free (non-protein bound) WN3 in time, using normal warfarin as internal standard. However, as they show in SI section 9 both warfarin and WN3 almost completely bind to plasma proteins within 4 hours. How can one determine the reactivity in plasma if ca 95-98 % of the compound is not taken into account due to the fact that it binds to protein and is removed before analysis. Apparently the authors assume that WN3 and warfarin have exactly the same binding kinetics to plasma protein. However, there is no basis for this.

Answer: To address this main concern and validate kinetic data, we have now performed additional experiments to demonstrate that acetonitrile extraction used throughout all kinetic analyses is an efficient method to recover all forms of warfarin derivatives from plasma. For this purpose, acetonitrile extractions were performed after incubation of Warfarin **1a** (internal standard), WN₃ **2** and clicked BCN-peg₆-OH product **10** in plasma (Figure S2 or clicked BCN product **A** for reviewers only: Figure R1). Please note that these analyses were done in the same conditions as those used for kinetic measurements. Each sample was then analysed by HPLC and compared with standard solution prepared by mixing the molecules (**1a**, **2**, **10** or **A**) in the same acetonitrile matrix obtained from normal plasma samples. No significant differences were observed between extracted and standard samples demonstrating that warfarin derivatives were completely extracted from plasma even if they are highly bound to plasmatic proteins (as indicated by Figure SI19-20). We added to SI these results obtained for

1a, 2 and 10 (SI, Figure S2) and at the end of this letter the results obtained for **1a, 2 and A** for reviewers information only (Figure R1).

The experiment in SI section 9 only looks at one time point and already gives a slightly different result for both compounds. The work in section 9 should be expanded to earlier time points to understand how fast and to what extent both compounds bind to proteins. In addition there the effect of concentration on plasma binding kinetics will need to be understood to know if the reaction kinetics in plasma can be trusted. Even a slight difference in binding, also at the earlier time points will result in a different k_2 . This in my opinion is the reason for the 10-fold higher k_2 in plasma compared to PBS. While I guess one can still use the current plasma k_2 to make a statement about stability in time (of the non bound fraction), it cannot be used as an argument that WN3 is equally reactive and available in plasma and PBS.

Answer: The acceleration of SPAAC kinetic reaction has been observed with non-modified BCN model compound. This strained alkyne has not been selected for click in vivo experiments owing to its poor hydrophilicity and the difficulty to detect it in plasma by MS/MS. The particular reactivity of this compound can easily be removed from our paper without weakening the message. Nevertheless, we also repeated the kinetic experiments using non-modified BCN for reviewers only (see Figure R1-R4). Taking into account the concerns of Reviewer 1 and 2 about kinetic of click reaction, we modified our method to monitor the appearance of clicked product. After demonstrating that warfarin derivatives can be completely extracted from plasma samples in acetonitrile (see above), we repeated three times all kinetic experiments by monitoring the appearance of the HPLC signal of clicked product (**10** or **A**) normalized against signal of internal standard, Warfarin **1a**.

Second-order rate constants for the reaction were then determined by plotting the $1/[WN_3]$ versus time and analysing by linear regression (**10**: Figure 7b, **A**: Figure R4b). Interestingly, the rate constants obtained with this new analytical method are comparable to those obtained with our initial method.

The authors attempt to address this by preincubating WN3 in plasma for 15 hours followed by assessing the reaction kinetics. The authors state that the found k_2 of 4.8 is similar to the earlier found 6.3 (without preincubation), but without error bars this statement cannot be made. It is still a ca 25 % drop and in addition the 15 hour time frame does not match the binding kinetics to protein, as in section 9 of the SI the authors show that within 4 hours ca 98 % of the WF3 has bound to plasma protein. As it stands now the paper only reports about the 2 % of WN3 that is not bound to protein, i.e. the soluble fractions. The protein bound fractions are removed before analysis; this also holds for the LCMS analysis of blood and liver samples. There is no evidence that protein bound WN3 reacts to BCN and that the product bound to protein is cleared. This should be made very clear to the reader. And as it stands now the k_2 in plasma cannot be presented as a k_2 in plasma.

Answer: We believe that our new additional data clearly demonstrate now that: i) molecules are quantitatively extracted from plasma samples, ii) kinetic measurement takes into account all (both plasma-protein-bound and free) WN_3 .

Regarding my earlier comment that the effect of the complete clearance of WN3 on anticoagulation appears to be modest (ca 2 fold):

The issues listed above seem to give ample room for the possibility that protein bound WN3 acts a reservoir and re-equilibrates the WN3 levels in blood after the click and clear.

Answer: We have now unambiguously shown that our plasmatic extraction procedure is reliable, we can thus reasonably exclude that the remaining anticoagulant activity is the result from re-equilibration from plasma since we did not detect WN_3 in the bloodstream 1h after neutralisation. As stated below, we cannot exclude latent re-equilibration of WN_3 from organs.

Based on Figure 5 C this does not seem to happen in the 1st hour, as the WN_3 levels are 0. However, the authors' statement (lines 372-375) that the modest effect on anticoagulation is the result of incomplete reaction of BCN with WN_3 is incorrect based on these results in Figure 5C, as the WN_3 clearance is complete.

Answer: We agree with this comment and discarded this sentence from the manuscript.

Therefore it is perhaps more likely that re-equilibration at later time points, between 1 and 5 h are the cause for the modest anticoagulation effect. At least this should be clearly addressed.

Answer: This assumption cannot be completely excluded as we have looked only at liver, blood and urine. It is thus possible that some re-equilibration may arise from other compartment at the time when the click reaction cannot proceed anymore due to the rapid elimination of the clearing agent *via* the urine. We thus propose to add in the text the following sentence:

"...These data suggest that the remaining low amount of WN₃ **2** in the liver after the click reaction account at least in part for the residual anticoagulant activity. However, we cannot exclude re-equilibration of WN₃ **2** from another compartment. Indeed, the neutralizing strained alkyne was designed to have a fast clearance preventing later reaction with re-equilibrated WN₃ **2**. We thus measured the renal clearance of WN₃ **2**, BCN-peg₆-OH **9** and clicked product **10**."

minor comments

line 137-8 " which is of the same order of magnitude than non-modified BCN (11.1 M⁻¹s⁻¹), while lower than in water." "lower" should be "higher"? Answer: This sentence has been removed.

Figure 7 caption, please add the time. Answer: Time was added in the Figure 7.

The grammar and spelling still needs attention, also in the SI. Answer: The manuscript has been proofread by an English native speaker to correct those mistakes and rephrase some sentences to clarify their meaning.

Additional experiments for reviewers:

Extraction and kinetic experiments using non-modified BCN have been added for reviewers only. These data have obtained using the same procedure described in SI for BCN-peg₆-OH **9** (see SI 4. Kinetic studies in human plasma).

Scheme R1 : SPAAC reaction between WN₃ (**2**) and non-modified BCN.

For the three compounds, the peak areas of extraction samples were not significantly different from those obtained for standard samples (Figure R1) demonstrating that warfarin-derivatives can be completely extracted from plasma using this procedure.

Figure R1 : Warfarin derivatives extraction from plasmatic samples. Blue: HPLC peak areas obtained for standard samples ([**1a**] = [**2**] = [**10**] = 20 μ M). Red: HPLC peak areas obtained for extraction samples (Theoretical concentration: [**1a**] = [**2**] = [**10**] = 20 μ M). These experiments have been repeated three times and error bars indicate standard deviation.

Calibration curves of clicked BCN product **A** in both PBS and acetonitrile were obtained (Figure R2) and used during kinetic studies to determine the concentration of clicked product **A**.

Figure R2 : Calibration curve of clicked BCN product (A) in PBS and ACN. a, Calibration curve of clicked BCN product **A** in PBS. b, Calibration curve of clicked BCN product **A** in ACN (d = 5).

Kinetic experiments were then recorded in PBS and in plasma at 100 μM (see SI for details). Corresponding rate constants of 0.8 (± 0.15) and 12.4 (± 1.5) $\text{M}^{-1}\text{s}^{-1}$ were found for BCN (see SI Figure R3 and R4).

Figure R3 : Kinetic experiments with WN_3 (2) and non-modified BCN in PBS. a, Conversion rate of clicked BCN product **A** versus time. b, Linear regression of $1/[\text{WN}_3 \mathbf{2}]$ versus time. These experiments have been repeated three times and error bars indicate standard deviation.

Figure R4 : Kinetic experiments with WN_3 (2**) and non-modified BCN in plasma.** a, Conversion rate of clicked BCN product **A** versus time. b, Linear regression of $1/[WN_3 \mathbf{2}]$ versus time. These experiments have been repeated three times and error bars indicate standard deviation.

Reviewer #2 (Remarks to the Author):

I was a previous reviewer of this manuscript. The revised submission is substantially improved. Many of my concerns have been addressed. (I am not qualified to comment on whether the other reviewer concerns were addressed.)

I still have some issues with the following:

1. How the rate/kinetic data are being interpreted. The authors note that the "click" reaction rate increases in plasma. (This is counterintuitive based on the mechanism of the reaction and what other groups have reported.) From what I can tell in the SI, the authors are using the disappearance of starting materials to measure rate constants. This is decidedly less desirable than monitoring product production. (There are many ways that the starting materials can be "consumed", including sequestration on hydrophobic proteins, etc. So, the rate constants being reported do not accurately reflect the ligation rates in the environments. The authors' claims on rapid reactivity (and even the amounts of reagents necessary for in vivo ligation) could be skewed by over-interpreted data.

Answer: As discussed above, kinetic experiments have been repeated taking now into account the appearance of clicked product:

"Taking account the concerns of Reviewer 1 and 2 about kinetic of click reaction, we modified our method to monitor the appearance of clicked product. After demonstrating that warfarin derivatives can be completely extracted from plasma samples in acetonitrile (see above), we repeated three times all kinetic experiments by monitoring the appearance of the HPLC signal of clicked product (**10** or **A**) normalized against signal of internal standard, Warfarin **1a**.

Second-order rate constants for the reaction were then determined by plotting the $1/[WN_3 \mathbf{2}]$ versus time and analysing by linear regression (**10**: Figure S7b, **A**: Figure R4b). Interestingly, the rate constants obtained with this new analytical method are comparable to those obtained with our initial method."

2. The presentation of reagent stability/robustness data. It is still not easy to see what the authors are claiming. Note in Figure S8 that no comparison data are provided to the non-heated/non-incubated controls. (Stability data are usually provided as half-life values or as "percent remaining"/normalized plots.)

Answer: To study the stability of $WN_3 \mathbf{2}$ and BCN-peg₆-OH **9**, we recorded kinetic experiments after pre-incubation in plasma (15h, 37°C) of one of these click reagents (see SI Figure S8 and S9) and directly compared the results obtained to data for fresh, non pre-incubated reagents (see SI Figure S7). This strategy allows not only to study reagents' stability in plasma but also to check their abilities to undergo click reaction after aging in plasma. The results are discussed in the manuscript and compared to the control experiment performed without reagent pre-incubation:

“WN₃ (2) and BCN-peg₆-OH (9) stability in plasma. WN₃ **2** was found to be stable when kept at 37°C in the plasma or as dried powder for more than 24 h without particular precaution. WN₃ **2** stability in plasma was also assessed by recording the kinetic of neutralization with BCN-peg₆-OH **9** after incubation of a solution of WN₃ **2** aged in plasma at 37°C for 15h. Under this condition a reaction rate constant of $5.1 \pm 0.8 \text{ M}^{-1}\text{s}^{-1}$ was measured (SI Figure S8b), close to that observed in the control experiment using fresh solution of reagents ($6.8 \pm 1.8 \text{ M}^{-1}\text{s}^{-1}$, SI Figure S7b).

BCN is known to be a reactive scaffold with limited stability in biological media. We thus decided to evaluate the effect of the time of residence of BCN-peg₆-OH **9** in the plasma at 37°C on its ability to neutralize WN₃ **2**. As expected, we found that after 15h of incubating BCN-peg₆-OH **9** in human plasma at 37°C, kinetic constant decreased with time from $6.8 \pm 1.8 \text{ M}^{-1}\text{s}^{-1}$ to $2.4 \pm 0.8 \text{ M}^{-1}\text{s}^{-1}$. This suggests that BCN-peg₆-OH **9** in plasma shows a suitable reactivity vs stability balance during limited time-window, yet sufficiently long to be suitable for *in vivo* experiments (compare Figure 7).”

Additional experiments for reviewers:

Extraction and kinetic experiments using non-modified BCN have been added for reviewers only. These data have obtained using the same procedure described in SI for BCN-peg₆-OH **9** (see SI 4. Kinetic studies in human plasma).

Scheme R1 : SPAAC reaction between WN₃ (**2**) and non-modified BCN.

For the three compounds, the peak areas of extraction samples were not significantly different from those obtained for standard samples (Figure R1) demonstrating that warfarin-derivatives can be completely extracted from plasma using this procedure.

Figure R1 : Warfarin derivatives extraction from plasmatic samples. Blue: HPLC peak areas obtained for standard samples ([**1a**] = [**2**] = [**10**] = 20 μ M). Red: HPLC peak areas obtained for extraction samples (Theoretical concentration: [**1a**] = [**2**] = [**10**] = 20 μ M). These experiments have been repeated three times and error bars indicate standard deviation.

Calibration curves of clicked BCN product **A** in both PBS and acetonitrile were obtained (Figure R2) and used during kinetic studies to determine the concentration of clicked product **A**.

Figure R2 : Calibration curve of clicked BCN product (A) in PBS and ACN. a, Calibration curve of clicked BCN product **A** in PBS. b, Calibration curve of clicked BCN product **A** in ACN (d = 5).

Kinetic experiments were then recorded in PBS and in plasma at 100 μM (see SI for details). Corresponding rate constants of 0.8 (± 0.15) and 12.4 (± 1.5) $\text{M}^{-1}\text{s}^{-1}$ were found for BCN (see SI Figure R3 and R4).

Figure R3 : Kinetic experiments with WN_3 (2) and non-modified BCN in PBS. a, Conversion rate of clicked BCN product **A** versus time. b, Linear regression of $1/[\text{WN}_3 \mathbf{2}]$ versus time. These experiments have been repeated three times and error bars indicate standard deviation.

Figure R4 : Kinetic experiments with WN_3 (2**) and non-modified BCN in plasma.** a, Conversion rate of clicked BCN product **A** versus time. b, Linear regression of $1/[WN_3 \mathbf{2}]$ versus time. These experiments have been repeated three times and error bars indicate standard deviation.

Reviewer #3 (Remarks to the Author):

The authors have addressed my major concerns. However, there are some minor points that need to be addressed to improve the flow of information.

1. For consistency, the authors should either show the prothrombin time ratios (as they do in Figure 4) or the prothrombin times (as they do in Figures 3 and 4), but not a mixture of both.

Answer: We have intentionally chosen to show two types of presentations as each type of presentation better reflects what we want to illustrate in the text. Thus Figure 3 and 4 show absolute values of prothrombin times (PT) to allow the reader a direct comparison of data in our experimental conditions with those reported previously. In the Figure 6, however, we show effects of combined treatments and our objective is to best illustrate the size of change of coagulating parameters obtained after injection of clearing agents or control treatment. We consider that showing ratios (or fold change) with respect to baseline values of control measures (equal 1) is the best method to illustrate the magnitude of clearing agent effect. For this reason, we would like to maintain present representation of data in figures 3,4 and 6.

2. The authors show the prothrombin times in factor II, VII or X deficient plasma before and after warfarin or WN3 administration (Figure S12) and after WN3 reversal with vitamin K or BCNpeg6OH (Figure S13). By comparison with a standard curve constructed by adding varying amounts of normal plasma, they need to use this information to determine the level of each of these clotting factors.

Answer: We have now established standard curves to quantify each factor with respect to human normal plasma, as suggested by the reviewer 3 (shown in new Figure S13) and calculated levels of factors for different treatments accordingly (see new Figure S14 and Figure S15). For the facility of future comparisons we have also indicated mean PT values obtained for each treatment (shown within corresponding bars of each graph).

3. *There still are numerous typographical and grammatical errors that require attention.*

Answer: The manuscript has been proofread by an English native speaker to correct those mistakes and rephrase some sentences to clarify their meaning.

Additional experiments for reviewers:

Extraction and kinetic experiments using non-modified BCN have been added for reviewers only. These data have obtained using the same procedure described in SI for BCN-peg₆-OH **9** (see SI 4. Kinetic studies in human plasma).

Scheme R1 : SPAAC reaction between WN₃ (**2**) and non-modified BCN.

For the three compounds, the peak areas of extraction samples were not significantly different from those obtained for standard samples (Figure R1) demonstrating that warfarin-derivatives can be completely extracted from plasma using this procedure.

Figure R1 : Warfarin derivatives extraction from plasmatic samples. Blue: HPLC peak areas obtained for standard samples ([**1a**] = [**2**] = [**10**] = 20 μ M). Red: HPLC peak areas obtained for extraction samples (Theoretical concentration: [**1a**] = [**2**] = [**10**] = 20 μ M). These experiments have been repeated three times and error bars indicate standard deviation.

Calibration curves of clicked BCN product **A** in both PBS and acetonitrile were obtained (Figure R2) and used during kinetic studies to determine the concentration of clicked product **A**.

Figure R2 : Calibration curve of clicked BCN product (A) in PBS and ACN. a, Calibration curve of clicked BCN product **A** in PBS. b, Calibration curve of clicked BCN product **A** in ACN (d = 5).

Kinetic experiments were then recorded in PBS and in plasma at 100 μM (see SI for details). Corresponding rate constants of 0.8 (± 0.15) and 12.4 (± 1.5) $\text{M}^{-1}\text{s}^{-1}$ were found for BCN (see SI Figure R3 and R4).

Figure R3 : Kinetic experiments with WN_3 (2) and non-modified BCN in PBS. a, Conversion rate of clicked BCN product **A** versus time. b, Linear regression of $1/[\text{WN}_3 \mathbf{2}]$ versus time. These experiments have been repeated three times and error bars indicate standard deviation.

Figure R4 : Kinetic experiments with WN_3 (2**) and non-modified BCN in plasma.** a, Conversion rate of clicked BCN product **A** versus time. b, Linear regression of $1/[WN_3 \mathbf{2}]$ versus time. These experiments have been repeated three times and error bars indicate standard deviation.

REVIEWERS' COMMENTS:

Reviewer #1 (Remarks to the Author):

The authors adequately addressed my previous comments and I have no further comments. I recommend publishing this paper and congratulate the authors on this nice work.

Reviewer #3 (Remarks to the Author):

The revised manuscript addresses my concerns.